# A multivariate Box-Behnken assessment of elevated branched-chain amino acid concentrations in reduced crude protein diets offered to male broiler chickens

**Peter V. Chrystal**[1,2], **Shiva Greenhalgh**[1,2], **Shemil P. Macelline**[1,2], **Juliano C. de Paula Dorigam**[3], **Peter H. Selle**[1], **Sonia Y. Liu**[1,2]*

**1** Poultry Research Foundation, Faculty of Science, The University of Sydney, Camden, NSW, Australia, **2** School of Life and Environmental Sciences, Faculty of Science, The University of Sydney, Sydney, NSW, Australia, **3** Evonik Operations GmbH, Hanau-Wolfgang, Germany

* sonia.liu@sydney.edu.au

## Abstract

In a Box-Behnken assessment of elevated branched-chain amino acids (BCAA), 13 nutritionally equivalent maize-based diets were offered to a total of 390 off-sex male Ross 308 broiler chickens from 7 to 28 days post-hatch. The BCAA concentrations investigated in reduced-crude protein diets were 12.5, 15.5, 18.3 g/kg leucine (125, 155, 183); 8.9, 10.2, 12.5 g/kg valine (89, 102, 125) and 7.2, 8.9, 10.8 g/kg isoleucine (72, 89, 109), where their relativity to 11.0 g/kg digestible lysine are shown in parentheses. Determined parameters included growth performance, relative abdominal fat-pad weights, nutrient utilisation, apparent digestibility coefficients, disappearance rates of 16 amino acids and free amino acid systemic plasma concentrations. Increasing dietary leucine linearly depressed weight gain and quadratically influenced FCR where the estimated minimum FCR of 1.418 was with 14.99 g/kg leucine. Polynomial regression analysis and surface response curves of determined parameters were generated for significant ($P < 0.05$) BCAA variables, based on lack of fit ($P > 0.005$). Quadratic and cross-product responses were observed for weight gain, FCR, AME, $AME_n$, N retention and apparent digestibility of 13 amino acids. Relative fat-pad weights declined linearly with increasing isoleucine and valine. The lowest N retention was estimated at a combination of 15.25 and 10.50 g/kg leucine and valine respectively whilst the highest mean digestibility coefficient (0.793) of amino acids was estimated at a combination of 15.74 and 10.47 g/kg of leucine and valine respectively. The remaining parameter minima or maxima responses were not able to be determined since they were outside the extreme BCAA treatment levels. Increasing dietary BCAA significantly increased apparent ileal digestibilities and disappearance rates of BCAA. Systemic plasma concentrations of valine increased ($P < 0.001$) with increasing dietary valine but leucine was not influenced ($P > 0.25$). Systemic plasma concentration of isoleucine was maximised ($P < 0.001$) only when accompanied by elevated dietary leucine. Also, dietary treatments influenced ($P < 0.05$) apparent disappearance rates of all the essential amino acids analysed, with the exception of methionine. Whilst overall growth performance was not disadvantaged ($P > 0.10$) by

**Data Availability Statement:** All relevant data are within the manuscript and its Supporting Information files.

**Funding:** The study is funded by Evonik Nutrition and Care GmbH, Hanau-Wolfgang 63457, Germany. The funder provided support in the form of salaries for authors Dr Juliano C. de P Dorigam, but did not have any additional role in the study design, data collection and analysis, decision to publish or preparation of the manuscript. The specific roles of these authors are articulated in the 'author contributions' section.

**Competing interests:** The study is funded by Evonik Nutrition and Care GmbH, Hanau-Wolfgang 63457, Germany. However, this does not alter our adherence to PLOS ONE policies on sharing data and materials. Dr Juliano C. de P Dorigam is employed by Evonik and he contributed to Funding acquisition and Project administration.

elevated BCAA levels, compared with 2019 Ross 308 performance objectives, polynomial regression analysis suggested both interaction and antagonism between BCAA.

## Introduction

The branched-chain amino acids (BCAA) leucine, valine and isoleucine account for 35% of the essential amino acids in muscle protein and 40% of the total amino acid requirement in poultry [1]. Furthermore, The BCAA regulate protein synthesis and turnover in broiler chickens, facilitate glucose uptake by skeletal muscle, enhance glycogen synthesis and are important regulators of mTOR signalling pathways [2,3]. Indeed, leucine was identified as a possible regulator of protein turnover in muscle decades ago [4] and has been shown to regulate both protein and lipid metabolism, promoting lean tissue gain in young animals and alleviating muscle protein loss in aging adults and food-deprived animals [5–8]. BCAA are actively degraded in extra-hepatic and extra-intestinal tissues and Wu [6] suggested dietary leucine, valine and isoleucine should be present in an appropriate ratio to prevent amino acid imbalance in meat-type chickens.

The antagonistic effect of disproportionate amounts of BCAA on broiler and rat performance has been well documented; excess leucine disrupts the utilisation of isoleucine and valine, especially when these two amino acids are marginal or limiting [9–14]. Leucine-induced BCAA antagonism was first observed in animals decades ago as the addition of 30 g/kg of *l*-leucine to a low-protein (90 g/kg casein) atypical diet caused growth depression in rats which could be partially overcome by supplementation of isoleucine [15]. Antagonistic BCAA interactions have been reported to alter BCAA concentrations in blood plasma and body tissues whilst high intake of leucine depressed overall bird performance [16] and the concentrations of valine and isoleucine in blood and muscle [15,17]. Moreover, for broiler chickens offered a valine deficient diet (6.3 g/kg) with adequate levels of isoleucine and leucine, poor weight gains were associated with a high incidence of feather and leg abnormalities [18]. However, birds offered diets adequate or deficient in total BCAA did not exhibit these signs and it was proposed by Farran [19] that the BCAA requirement for broilers may be influenced by BCAA antagonism. In contrast to these observations, recently Zeitz et al. [20,21] investigated elevated dietary leucine by 35 and 60% above the breeder recommendations in the initial study (Ross 308) and AMINO*Dat*® 5.0 in a subsequent study (Cobb 500). These authors maintained constant isoleucine:valine ratios in the initial study and increased isoleucine and valine in tandem with leucine in the second study. Conclusions were similar; elevating dietary leucine by 60% did not influence molecular pathways of protein synthesis or degradation and did not affect growth performance of rapid growing male broilers. However, whilst antagonism between the BCAA is unlikely to result in depressed broiler performance when practical type diets are offered, this may not be the case with reduced-CP diets [22].

Maximum weight gain and feed conversion efficiency in male broilers to 21 days post-hatch offered dietary levels of 11.6 g/kg leucine, 9.0 g/kg valine and 7.8 g/kg isoleucine have been proposed [18]. Instructively, Ospina-Rojas et al. [1] comprehensively investigated responses of BCAA in Cobb 500 male broilers offered reduced crude protein from 21 to 42 days post-hatch (160 g/kg crude protein and 10.4 g/kg digestible lysine). The ratio of digestible leucine to digestible lysine (100) ranged from 96 to 173 and the ratio of digestible valine ranged from 50 to 108 whilst isoleucine was kept constant at 7.1 g/kg. Increasing dietary digestible leucine resulted in linear reductions in feed intake, weight gain and feed conversion efficiency. In

contrast, increasing dietary digestible valine resulted in quadratic improvements in these performance parameters. Additionally, interactions between digestible leucine × digestible valine for weight gain and feed intake but not feed conversion efficiency were observed. Interestingly, there was a linear decline in breast meat yield and abdominal lipid with increasing dietary digestible leucine. Furthemore, increasing dietary digestible valine resulted in a quadratic increase in thigh yield and a linear decline in abdominal lipid percent.

In 2008, *l*-valine became commercially available followed by *l*-isoleucine [23] allowing supplementation of reduced crude protein diets with BCAA. The removal of supplemented Val from a balanced reduced crude protein diet caused the largest reduction on growth performance in comparison to removal of others supplemented amino acids and the removal of Leu significantly increased Val and Ile concentrations in plasma [24]. Previous studies examined moderate ranges of BCAA derived from formulating conventional diets [25,26] and it is hypothesised the importance of BCAA may be more pronouced in reduced CP diets, hence the purpose of the current study was to investigate BCAA antagonism in male broilers by varying elevated levels of all three BCAA, utilising a multivariate Box-Behnken experimental design with tangibly reduced crude protein diets and amino acid ratios meeting or exceeding the ideal protein concept of the primary breeder (Aviagen Ross 308, 2019). The advantage of the Box-Behnken design is that multiple nutrient changes may be studied simultaneously whilst the number of treatment combinations is substantially reduced compared with a full factorial design [27]. Furthermore, it may be used as a tool to predict a desired response and thus determine the factors that optimise this response.

## Materials and methods

### Experimental design

A 3-factor, 3-level Box-Behnken design was utilised to determine the impact of leucine, valine and isoleucine (Table 1) on growth performance and carcass parameters of Ross 308 off-sex male broilers from 7 to 28 days post-hatch. The present study consisted of 13 dietary treatments as outlined in Table 2, where energy density, crude protein, standardised ileal digestible lysine and dietary electrolyte balance (DEB) were maintained constant across all diets at 12.97 MJ/kg, 175 g/kg, 11.00 g/kg and 230 mEq/kg, respectively. The diets were formulated on the basis of near-infrared spectroscopy (NIRS; AMINO*Nir*® Advanced, Evonik Operations GmbH. Hanau, Germany) of maize, soybean meal and canola seed. The values of the centre points for the BCAA were 15.43, 10.53 and 9.00 g/kg for leucine, valine and isoleucine respectively as described in treatment 13. The evaluated dependant variable response for weight gain, feed intake, feed conversion ratio (FCR) mortality, relative fat pad weights, nutrient utilisation, apparent metabolisable energy, apparent amino acid digestibility, and systemic blood plasma amino acid content were determined.

**Table 1. Relativity of three tiers of digestible branched-chain amino acid concentrations to lysine (100) in experimental diets[1] containing 11.0 g/kg digestible lysine and 175 g/kg crude protein applied to a 3-factor, 3-level Box-Behnken design.**

| Amino acid factors | Level (-) | Level (0) | Level (+) |
|---|---|---|---|
| $X_1$: Leucine | 125 | 155 | 183 |
| $X_2$: Valine | 89 | 102 | 125 |
| $X_3$: Isoleucine | 72 | 89 | 109 |

[1]The thirteen experimental diets contained an average of 34.10 g/kg unbound amino acids ranging from 26.25 to 41.56 g/kg.

**Table 2. List of experimental treatments for Ross 308 off-sex male broiler chickens from 7 to 28 days post-hatch.**

| Treatment | Code | Leucine (g/kg) | Valine (g/kg) | Isoleucine (g/kg) |
|---|---|---|---|---|
| 1A | 0 + + | 15.50 | 12.50 | 10.90 |
| 2B | 0 − + | 15.50 | 8.90 | 10.90 |
| 3C | 0 + − | 15.50 | 12.50 | 7.20 |
| 4D | 0 − − | 15.50 | 8.90 | 7.20 |
| 5E | + + 0 | 18.30 | 12.50 | 8.90 |
| 6F | − + 0 | 12.50 | 12.50 | 8.90 |
| 7G | + − 0 | 18.30 | 8.90 | 8.90 |
| 8H | − − 0 | 12.50 | 8.90 | 8.90 |
| 9I | + 0 + | 18.30 | 10.20 | 10.90 |
| 10J | − 0 + | 12.50 | 10.20 | 10.90 |
| 11K | + 0 − | 18.30 | 10.20 | 7.20 |
| 12L | − 0 − | 12.50 | 10.20 | 7.20 |
| 13M | centre | 15.43 | 10.53 | 9.00 |

## Diet preparation

The composition and calculated nutrient specifications of the experimental maize/soybean meal/canola seed-based diets are shown in Tables 3 and 4. Analysed crude protein and amino acid concentrations in Table 5. Maize was coarsely ground (6.0 mm hammer-mill screen)

**Table 3. Composition of experimental diets (g/kg).**

| Ingredients/Treatment | 1A | 2B | 3C | 4D | 5E | 6F | 7G | 8H | 9I | 10J | 11K | 12L | 13M |
|---|---|---|---|---|---|---|---|---|---|---|---|---|---|
| Maize | 675 | 667 | 668 | 659 | 677 | 665 | 669 | 656 | 676 | 664 | 668 | 656 | 667 |
| Soybean meal | 161 | 173 | 172 | 184 | 159 | 176 | 171 | 187 | 161 | 178 | 171 | 188 | 173 |
| Canola seed | 60.0 | 60.0 | 60.0 | 60.0 | 60.0 | 60.0 | 60.0 | 60.0 | 60.0 | 60.0 | 60.0 | 60.0 | 60.0 |
| Soy oil | 3.48 | 6.56 | 6.87 | 9.95 | 2.77 | 8.03 | 5.85 | 11.1 | 2.91 | 8.17 | 6.29 | 11.6 | 6.96 |
| l-lysine HCl | 5.90 | 5.55 | 5.59 | 5.24 | 5.97 | 5.47 | 5.62 | 5.12 | 5.92 | 5.42 | 5.61 | 5.10 | 5.54 |
| d,l-methionine | 3.95 | 3.85 | 3.86 | 3.75 | 3.97 | 3.82 | 3.87 | 3.72 | 3.96 | 3.81 | 3.86 | 3.71 | 3.84 |
| l-threonine | 2.93 | 2.77 | 2.78 | 2.62 | 2.96 | 2.73 | 2.80 | 2.57 | 2.94 | 2.71 | 2.79 | 2.56 | 2.76 |
| l-tryptophan | 0.39 | 0.32 | 0.33 | 0.27 | 0.40 | 0.31 | 0.34 | 0.25 | 0.39 | 0.30 | 0.33 | 0.24 | 0.32 |
| l-valine | 6.74 | 2.87 | 6.56 | 2.69 | 6.79 | 6.50 | 2.91 | 2.62 | 4.41 | 4.12 | 4.23 | 3.94 | 4.53 |
| l-arginine | 3.40 | 3.07 | 3.10 | 2.76 | 3.47 | 2.99 | 3.14 | 2.65 | 3.42 | 2.94 | 3.12 | 2.63 | 3.06 |
| l-isoleucine | 5.97 | 5.77 | 2.02 | 1.82 | 3.97 | 3.69 | 3.77 | 3.49 | 5.98 | 5.70 | 2.03 | 1.74 | 3.83 |
| l-leucine | 4.84 | 4.53 | 4.56 | 4.25 | 7.74 | 1.41 | 7.44 | 1.10 | 7.70 | 1.37 | 7.42 | 1.09 | 4.45 |
| l-histidine | 0.28 | 0.17 | 0.18 | 0.07 | 0.31 | 0.15 | 0.20 | 0.04 | 0.29 | 0.13 | 0.19 | 0.03 | 0.17 |
| Glycine | 5.91 | 5.59 | 5.63 | 5.31 | 5.98 | 5.52 | 5.66 | 5.20 | 5.93 | 5.47 | 5.64 | 5.19 | 5.59 |
| Sodium bicarbonate | 7.24 | 7.23 | 7.23 | 7.22 | 7.24 | 7.23 | 7.23 | 7.22 | 7.24 | 7.23 | 7.23 | 7.22 | 7.23 |
| Potassium carbonate | 2.04 | 1.51 | 1.56 | 1.03 | 2.15 | 1.39 | 1.62 | 0.86 | 2.07 | 1.31 | 1.59 | 0.83 | 1.50 |
| Limestone | 11.2 | 11.1 | 11.1 | 11.1 | 11.2 | 11.1 | 11.2 | 11.1 | 11.2 | 11.1 | 11.1 | 11.1 | 11.1 |
| Dicalcium phosphate | 16.4 | 16.2 | 16.3 | 16.1 | 16.4 | 16.2 | 16.3 | 16.1 | 16.4 | 16.2 | 16.3 | 16.1 | 16.2 |
| Choline chloride (60%) | 0.90 | 0.90 | 0.90 | 0.90 | 0.90 | 0.90 | 0.90 | 0.90 | 0.90 | 0.90 | 0.90 | 0.90 | 0.90 |
| Celite | 20.0 | 20.0 | 20.0 | 20.0 | 20.0 | 20.0 | 20.0 | 20.0 | 20.0 | 20.0 | 20.0 | 20.0 | 20.0 |
| Vitamin-mineral premix[1] | 2.00 | 2.00 | 2.00 | 2.00 | 2.00 | 2.00 | 2.00 | 2.00 | 2.00 | 2.00 | 2.00 | 2.00 | 2.00 |

[1]Vitamin-trace mineral premix supplies in MIU/kg or mg/kg of diet: [MIU] retinol 12, cholecalciferol 5, [mg] tocopherol 50, menadione 3, thiamine 3, riboflavin 9, pyridoxine 5, cobalamin 0.025, niacin 50, pantothenate 18, folate 2, biotin 0.2, copper 20, iron 40, manganese 110, cobalt 0.25, iodine 1, molybdenum 2, zinc 90, selenium 0.3.

**Table 4. Nutrient specifications (g/kg) of experimental diets where values for digestible amino acids are tabulated.**

| Nutrient/Treatments | 1A | 2B | 3C | 4D | 5E | 6F | 7G | 8H | 9I | 10J | 11K | 12L | 13M |
|---|---|---|---|---|---|---|---|---|---|---|---|---|---|
| Metabolisable energy (MJ/kg) | 12.97 | 12.97 | 12.97 | 12.97 | 12.97 | 12.97 | 12.97 | 12.97 | 12.97 | 12.97 | 12.97 | 12.97 | 12.97 |
| Crude protein | 175 | 175 | 175 | 175 | 175 | 175 | 175 | 175 | 175 | 175 | 175 | 175 | 175 |
| Leucine | 15.50 | 15.50 | 15.50 | 15.50 | 18.30 | 12.50 | 18.30 | 12.50 | 18.30 | 12.50 | 18.30 | 12.50 | 15.43 |
| Valine | 12.50 | 8.90 | 12.50 | 8.90 | 12.50 | 12.50 | 8.90 | 8.90 | 10.20 | 10.20 | 10.20 | 10.20 | 10.53 |
| Isoleucine | 10.90 | 10.90 | 7.20 | 7.20 | 8.90 | 8.90 | 8.90 | 8.90 | 10.90 | 10.90 | 7.20 | 7.20 | 9.00 |
| Lysine | 11.00 | 11.00 | 11.00 | 11.00 | 11.00 | 11.00 | 11.00 | 11.00 | 11.00 | 11.00 | 11.00 | 11.00 | 11.00 |
| Methionine | 5.98 | 5.93 | 5.94 | 5.89 | 5.99 | 5.92 | 5.94 | 5.87 | 5.98 | 5.91 | 5.94 | 5.87 | 5.93 |
| Methionine + cysteine | 8.14 | 8.14 | 8.14 | 8.14 | 8.14 | 8.14 | 8.14 | 8.14 | 8.14 | 8.14 | 8.14 | 8.14 | 8.14 |
| Threonine | 7.37 | 7.37 | 7.37 | 7.37 | 7.37 | 7.37 | 7.37 | 7.37 | 7.37 | 7.37 | 7.37 | 7.37 | 7.37 |
| Tryptophan | 1.82 | 1.82 | 1.82 | 1.82 | 1.82 | 1.82 | 1.82 | 1.82 | 1.82 | 1.82 | 1.82 | 1.82 | 1.82 |
| Arginine | 11.44 | 11.44 | 11.44 | 11.44 | 11.44 | 11.44 | 11.44 | 11.44 | 11.44 | 11.44 | 11.44 | 11.44 | 11.44 |
| Histidine | 3.63 | 3.63 | 3.63 | 3.63 | 3.63 | 3.63 | 3.63 | 3.63 | 3.63 | 3.63 | 3.63 | 3.63 | 3.63 |
| Phenylalanine | 6.06 | 6.06 | 6.06 | 6.06 | 6.06 | 6.06 | 6.06 | 6.06 | 6.06 | 6.06 | 6.06 | 6.06 | 6.06 |
| Glycine[1] | 10.72 | 10.57 | 10.59 | 10.44 | 10.75 | 10.54 | 10.60 | 10.40 | 10.73 | 10.52 | 10.60 | 10.39 | 10.57 |
| Serine[1] | 6.00 | 6.20 | 6.18 | 6.38 | 5.96 | 6.25 | 6.16 | 6.45 | 5.99 | 6.28 | 6.17 | 6.46 | 6.21 |
| Calcium | 8.25 | 8.25 | 8.25 | 8.25 | 8.25 | 8.25 | 8.25 | 8.25 | 8.25 | 8.25 | 8.25 | 8.25 | 8.25 |
| Total phosphorus | 6.06 | 6.09 | 6.09 | 6.12 | 6.06 | 6.10 | 6.09 | 6.13 | 6.06 | 6.10 | 6.09 | 6.13 | 6.09 |
| Available phosphorus | 4.13 | 4.13 | 4.13 | 4.13 | 4.13 | 4.13 | 4.13 | 4.13 | 4.13 | 4.13 | 4.13 | 4.13 | 4.13 |
| Sodium[2] | 2.00 | 2.00 | 2.00 | 2.00 | 2.00 | 2.00 | 2.00 | 2.00 | 2.00 | 2.00 | 2.00 | 2.00 | 2.00 |
| Potassium[2] | 7.39 | 7.32 | 7.33 | 7.25 | 7.40 | 7.30 | 7.33 | 7.23 | 7.39 | 7.29 | 7.33 | 7.23 | 7.32 |
| Chloride[2] | 1.63 | 1.56 | 1.57 | 1.51 | 1.64 | 1.55 | 1.58 | 1.49 | 1.63 | 1.54 | 1.57 | 1.48 | 1.56 |
| Starch | 432 | 427 | 427 | 422 | 433 | 425 | 428 | 420 | 432 | 424 | 427 | 420 | 426 |
| Crude fat | 59.7 | 62.6 | 62.9 | 65.9 | 59.0 | 64.1 | 62.0 | 67.0 | 59.1 | 64.2 | 62.4 | 67.4 | 63.0 |
| Crude fibre | 26.2 | 26.3 | 26.3 | 26.4 | 26.2 | 26.3 | 26.3 | 26.4 | 26.2 | 26.3 | 26.3 | 26.4 | 26.3 |
| Leucine:lysine (100) | 141 | 141 | 141 | 141 | 166 | 114 | 166 | 114 | 166 | 114 | 166 | 114 | 140 |
| Valine:lysine (100) | 114 | 81 | 114 | 81 | 114 | 114 | 81 | 81 | 93 | 93 | 93 | 93 | 96 |
| Isoleucine:lysine (100) | 99 | 99 | 65 | 65 | 81 | 81 | 81 | 81 | 99 | 99 | 65 | 65 | 82 |

[1]All diets formulated to contain 15.00 g/kg glycine equivalents.

[2]All diets formulated to a dietary electrolyte balance of 230 mEq/kg.

prior to incorporation into the complete diets. All diets were steam-pelleted at a temperature of 80˚C with a residence time of 14 seconds in the conditioner using a Palmer PP330 pellet press (Palmer Milling Engineering, Griffith, NSW, Australia). Acid insoluble ash (Celite®, Celite Corporation. Lompoc, California) was included in diets at 20 g/kg as an inert dietary marker to determine N and amino acid digestibility coefficients.

## Bird management

This feeding study was approved by the Research Integrity and Ethics Administration of The University of Sydney (Project number 2016/973). A total of 390 off-sex, male Ross 308 chicks were procured from a commercial hatchery and were initially offered a standard Aviagen Ross 308 (2019) starter diet (239 g/kg crude protein, 12.8 g/kg digestible lysine and 12.55 MJ/kg AME). At 7 days post-hatch, birds were individually identified (wing-tags) and allocated into bioassay cages on the basis of body-weight so that there were no statistical difference in mean body-weights between cages. Each of the dietary treatments was offered to five replicate cages (6 birds per cage) from 7 to 28 days post-hatch. Broilers had unlimited access to water and feed and was under 23 hours illumination for the first three days followed by 16 hours

**Table 5. Analysed concentrations of protein (N) and amino acids in experimental diets.**

| Ingredient (g/kg) | 1A | 2B | 3C | 4D | 5E | 6F | 7G | 8H | 9I | 10J | 11K | 12L | 13M |
|---|---|---|---|---|---|---|---|---|---|---|---|---|---|
| Protein (N) | 174 | 175 | 179 | 173 | 175 | 170 | 170 | 170 | 172 | 180 | 180 | 177 | 164 |
| Arginine | 11.9 | 11.7 | 11.9 | 11.8 | 11.9 | 11.4 | 10.7 | 10.7 | 11.1 | 12.4 | 12.6 | 12.1 | 11.5 |
| Histidine | 4.1 | 4.0 | 4.1 | 4.0 | 4.1 | 3.9 | 4.0 | 3.9 | 4.0 | 4.2 | 4.2 | 4.2 | 4.0 |
| Isoleucine | 10.1 | 10.6 | 8.0 | 7.9 | 9.3 | 9.6 | 8.2 | 8.9 | 10.2 | 11.0 | 8.3 | 8.2 | 8.8 |
| Leucine | 16.9 | 17.3 | 17.3 | 17.3 | 19.5 | 14.5 | 19.2 | 14.9 | 19.3 | 15.4 | 20.2 | 15.2 | 16.8 |
| Lysine | 11.4 | 11.4 | 11.6 | 11.5 | 11.7 | 10.9 | 11.1 | 11.4 | 11.3 | 12.0 | 12.0 | 11.9 | 10.8 |
| Methionine | 5.6 | 5.6 | 5.7 | 5.4 | 5.5 | 5.6 | 5.5 | 5.6 | 5.5 | 5.7 | 5.8 | 5.7 | 5.3 |
| Phenylalanine | 7.0 | 7.4 | 7.3 | 7.5 | 7.0 | 7.0 | 7.1 | 7.5 | 7.0 | 7.6 | 7.5 | 7.6 | 7.0 |
| Threonine | 7.7 | 7.8 | 7.8 | 7.8 | 7.7 | 7.6 | 7.7 | 7.7 | 7.7 | 8.1 | 8.2 | 8.2 | 7.7 |
| Valine | 12.8 | 9.9 | 12.9 | 10.0 | 13.0 | 12.2 | 9.8 | 9.7 | 10.7 | 11.2 | 11.5 | 11.2 | 10.6 |
| Alanine | 7.9 | 8.2 | 8.1 | 8.3 | 7.9 | 8.1 | 8.0 | 8.2 | 8.0 | 8.5 | 8.4 | 8.6 | 8.1 |
| Aspartic acid | 13.2 | 13.7 | 13.8 | 14.2 | 13.3 | 13.3 | 13.2 | 14.1 | 13.3 | 14.7 | 14.8 | 15.0 | 13.4 |
| Glutamic acid | 25.5 | 26.6 | 26.5 | 27.3 | 25.8 | 25.8 | 25.8 | 27.0 | 25.8 | 28.0 | 28.2 | 28.3 | 25.9 |
| Glycine | 11.2 | 10.8 | 11.3 | 11.0 | 11.3 | 10.4 | 10.7 | 10.8 | 10.9 | 11.2 | 11.4 | 11.3 | 10.3 |
| Proline | 9.4 | 9.8 | 10.0 | 9.9 | 9.4 | 9.2 | 9.5 | 9.8 | 9.3 | 9.8 | 9.8 | 9.5 | 9.4 |
| Serine | 6.5 | 6.8 | 6.8 | 7.0 | 6.5 | 6.7 | 6.6 | 6.9 | 6.7 | 7.5 | 7.4 | 7.5 | 7.0 |
| Tryptophan | 1.9 | 1.9 | 2.0 | 1.9 | 1.9 | _[1] | 1.9 | 1.9 | 1.9 | 2.0 | 2.0 | 2.0 | 1.8 |

[1]Analysis not supplied.

illumination for the remainder of the study. There was an initial room temperature of 32°C, which was gradually decreased to 22°C by the end of the feeding study. Body weights and feed intakes were monitored from which feed conversion ratios (FCR) were calculated. The incidence of dead or culled birds was recorded daily and their body-weights used to adjust feed intakes per cage and correct FCR calculations.

## Sample collection and chemical analysis

Total excreta was collected from 22 to 24 days post-hatch from each cage to determine parameters of nutrient utilisation, including apparent metabolisable energy (AME), metabolisable energy to gross energy ratios (ME:GE), nitrogen (N) retention and N-corrected apparent metabolisable energy ($AME_n$). Excreta was dried in a forced-air oven at 80°C for 24 hours following collection. The GE of diets and excreta was determined by bomb calorimetry using an adiabatic calorimeter (Parr 1281 bomb calorimeter, Parr Instruments Co., Moline, IL).

The AME values were calculated on a dry matter basis from the following equation:

$$AME_{diet} = \frac{(Feed\ intake \times GE_{diet}) - (Excreta\ output \times GE_{excreta})}{(Feed\ intake)}$$

ME:GE ratios were calculated by dividing AME by the GE of the appropriate diets. N contents of diets and excreta were determined using a nitrogen determinator (Leco Corporation, St Joseph, MI) and N retention calculated from the following equation:

$$N\ Retention\ (\%) = \frac{(Feed\ intake \times Nutrient_{diet}) - (Excreta\ output \times Nutrient_{excreta})}{(Feed\ intake \times Nutrient_{diet})} \times 100$$

N-corrected AME values on a dry matter basis were calculated by correcting N retention to zero using the factor of 36.54 kJ/g N retained in the body [28].

The removal of supplemented Leu increased Val and Ile levels in plasma; whereas the removal of Val and Ile individually did not alter BCAA concentrations in plasma [24]. Therefore, at 27 days post-hatch, three birds at random were selected from each cage of the highest leucine treatments (5E, 7G, 9I and 11K) and systemic blood samples were taken from the brachial (wing) vein. Blood samples were then pooled, centrifuged and the decanted plasma samples were then kept at −80°C before analysis. Concentrations of twenty proteinogenic amino acids in plasma taken from the brachial vein were determined using precolumn derivatisation amino acid analysis with 6-aminoquinolyl-N-hydroxysuccinimidyl carbamate (AQC; Waters™ AccQTag Ultra) followed by separation of the derivatives and quantification by reversed phase ultra-performance liquid chromatography [29].

At day 28 post-hatch, birds were euthanased by intravenous sodium pentobarbitone injection. The small intestine was removed and digesta samples were collected in their entirety from the distal jejunum and distal ileum. The distal jejunum was demarcated by the mid-point between the end of the duodenal loop and Meckel's diverticulum and the distal ileum by the mid-point between Meckel's diverticulum and the ileo-caecal junction. Digesta was taken from the posterior portion of each small intestinal segment. Digesta samples from each cage were pooled, homogenized, freeze-dried and ground through 0.5 mm screen. The samples were then analysed for concentrations of protein and amino acids. Protein (N) concentrations were determined as outlined by Siriwan et al. [30].

Amino acid concentrations of diets and freeze-dried digesta were analysed by an amino acid analyser according to AOAC (1990) methodology (994.12). Plasma amino acid concentrations determined following 24-hour liquid hydrolysis at 110°C in 6 M HCl and then sixteen amino acids are analysed using the Waters AccQTag Ultra chemistry on a Waters Acquity UPLC. Acid insoluble ash (AIA) was included in the diets at 20 g/kg as an inert marker and apparent digestibility coefficients (ADC) of protein (N) and amino acids were calculated by the following equation:

$$\text{ADC} = \frac{(\text{Nutrient}/\text{AIA})_{\text{diet}} - (\text{Nutrient}/\text{AIA})_{\text{digesta}}}{(\text{Nutrient}/\text{AIA})_{\text{diet}}}$$

Protein (N) disappearance rates (g/bird/day) were deduced from feed intakes over the final phase of the feeding period from the following equation:

$$\text{Disappearance rate} = \text{feed intake (g/bird)} \times \text{dietary nutrient (g/kg)} \times \text{ADC}$$

## Statistical analysis

A completely randomised three level (−, 0, +) three factor (leucine, valine, isoleucine) Box-Behnken multivariate design was employed with five replicates per dietary treatment. This design allows multiple variable testing without constructing full factorials, thereby reducing the required number of treatments and facilitating generation of relevant surface plots [24]. Experimental data were analysed applying response surface methodology (RSM) in JMP® Pro 15.2.0 (SAS Institute Inc. JMP Software. Cary, NC, 2019). Linear and quadratic regressions for independent variables and interactions among variables were performed. Variables found significant ($P < 0.05$) were retained in the model and polynomial regression equations were generated as follows:

$$Y_n = b_0 + b_1X_1 + b_2X_2 + b_3X_3 + b_4X_1^2 + b_5X_2^2 + b_6X_3^2 + b_7X_1X_2 + b_8X_1X_3 + b_9X_2X_3$$

Where $Y_n$ is the estimated response of the dependant variable, $b_0$ is the intercept and $b_1$ to $b_9$ are the potential coefficients for the polynomial regression. Experimental units were the

**Table 6. Effects of dietary treatments on growth performance and relative abdominal fat-pad weights from 7 to 28 days post-hatch.**

| Dietary treatment | | | | Growth performance | | | | Relative abdominal fat-pad weights (g/kg) |
|---|---|---|---|---|---|---|---|---|
| Leucine (Lys ratio) | Valine (Lys ratio) | Isoleucine (Lys ratio) | Diet | Weight gain (g/bird) | Feed intake (g/bird) | FCR (g/g) | Mortality rate (%) | |
| 141 | 114 | 99.1 | 1 | 1403 | 2004 | 1.428 | 0.00 | 13.13 |
| 141 | 80.9 | 99.1 | 2 | 1453 | 2032 | 1.399 | 0.00 | 12.27 |
| 141 | 114 | 65.5 | 3 | 1468 | 2069 | 1.409 | 3.33 | 12.97 |
| 141 | 80.9 | 65.5 | 4 | 1408 | 2013 | 1.430 | 3.33 | 14.09 |
| 166 | 114 | 80.9 | 5 | 1410 | 2027 | 1.438 | 6.67 | 12.70 |
| 114 | 114 | 80.9 | 6 | 1385 | 2010 | 1.452 | 0.00 | 12.56 |
| 166 | 80.9 | 80.9 | 7 | 1401 | 2027 | 1.448 | 0.00 | 13.57 |
| 114 | 80.9 | 80.9 | 8 | 1469 | 2089 | 1.422 | 0.00 | 13.63 |
| 166 | 92.7 | 99.1 | 9 | 1431 | 2056 | 1.437 | 0.00 | 13.11 |
| 114 | 92.7 | 99.1 | 10 | 1436 | 2052 | 1.429 | 0.00 | 12.61 |
| 166 | 92.7 | 65.5 | 11 | 1403 | 2034 | 1.450 | 6.67 | 14.20 |
| 114 | 92.7 | 65.5 | 12 | 1447 | 2062 | 1.426 | 0.00 | 13.60 |
| 140 | 95.8 | 81.8 | 13 | 1433 | 2039 | 1.423 | 0.00 | 12.60 |
| SEM | | | | 25.18 | 34.64 | 0.0129 | 2.068 | 0.565 |

cage means and response surface plots were generated for statistically significant first, second degree and cross-product polynomial regressions. The optimal response of the dependant variables were calculated from these equations where relevant.

## Results

The effects of dietary treatments on growth performance, mortality rates and relative abdominal fat-pad weights are shown in Table 6 and the effect of dietary treatments on parameters of nutrient utilisation are displayed in Table 7. Independent of BCAA ratios, increasing digestible leucine levels linearly decreased (P = 0.046; r = – 0.358) weight gain, as shown in Fig 1. There was a quadratic response (P = 0.014; r = 0.358) in FCR to increasing dietary digestible leucine

**Table 7. Effects of dietary treatments on nutrient utilisation from 24 to 27 days post-hatch.**

| Dietary treatment | | | | Nutrient utilisation | | | |
|---|---|---|---|---|---|---|---|
| Leucine (Lys ratio) | Valine (Lys ratio) | Isoleucine (Lys ratio) | Diet | AME (MJ/kg DM) | ME:GE ratio (MJ/MJ) | N retention (%) | AME$_n$ (MJ/kg DM) |
| 141 | 114 | 99.1 | 1 | 13.33 | 0.818 | 75.38 | 12.28 |
| 141 | 80.9 | 99.1 | 2 | 13.07 | 0.807 | 72.86 | 12.09 |
| 141 | 114 | 65.5 | 3 | 13.44 | 0.821 | 74.46 | 12.43 |
| 141 | 80.9 | 65.5 | 4 | 13.48 | 0.817 | 71.97 | 12.56 |
| 166 | 114 | 80.9 | 5 | 13.39 | 0.821 | 75.32 | 12.38 |
| 114 | 114 | 80.9 | 6 | 13.26 | 0.814 | 74.18 | 12.25 |
| 166 | 80.9 | 80.9 | 7 | 13.49 | 0.823 | 75.28 | 12.43 |
| 114 | 80.9 | 80.9 | 8 | 13.33 | 0.823 | 76.47 | 12.19 |
| 166 | 92.7 | 99.1 | 9 | 13.16 | 0.814 | 74.68 | 12.09 |
| 114 | 92.7 | 99.1 | 10 | 13.28 | 0.817 | 73.34 | 12.26 |
| 166 | 92.7 | 65.5 | 11 | 13.33 | 0.818 | 73.74 | 12.34 |
| 114 | 92.7 | 65.5 | 12 | 13.37 | 0.820 | 73.59 | 12.39 |
| 140 | 95.8 | 81.8 | 13 | 13.35 | 0.819 | 71.98 | 12.41 |
| SEM | | | | 0.0646 | 0.0039 | 0.7812 | 0.0767 |

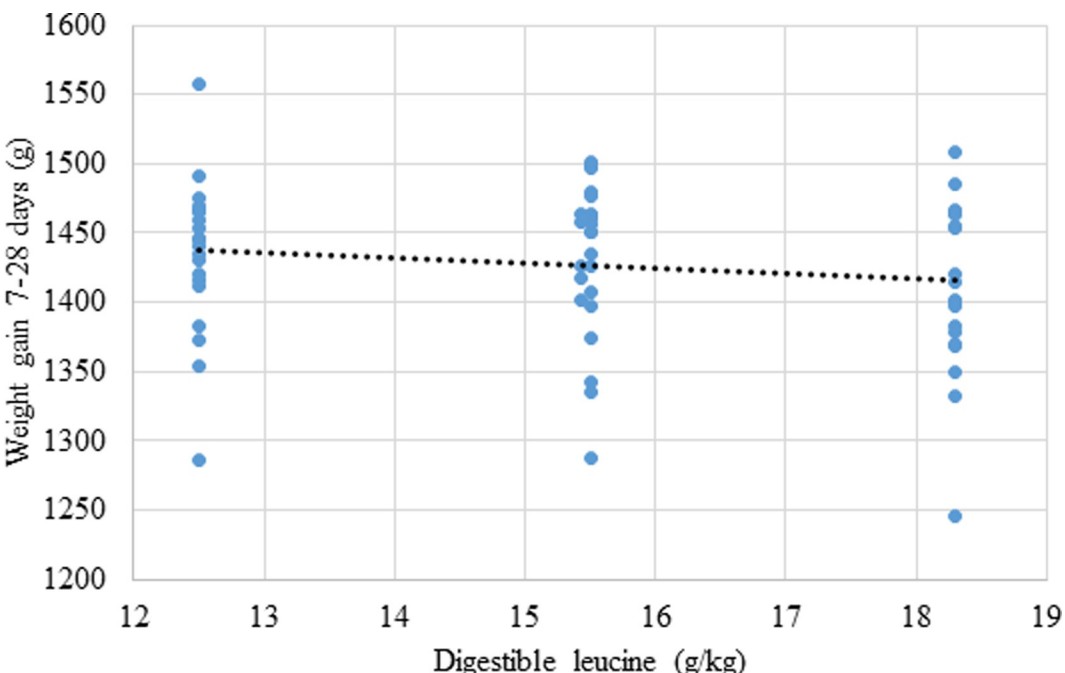

**Fig 1. Linear relationship (P = 0.022; r = -0.284) between dietary leucine concentrations and weight gain where: Weight gain$_{(g, 7-28d)}$ = 1487–3.905 × digestible leucine$_{(g/kg)}$).**

relative to lysine where the predicted minimum FCR of 1.418 corresponded to a dietary digestible leucine concentration of 14.99 g/kg as shown in Fig 2. Also, relative fat pad weights decreased linearly (P = 0.022; r = −0.284) with increasing dietary isoleucine (Fig 3).

Polynomial regression analysis for quadratic and cross-product differences (P < 0.05) based on lack of fit (P > 0.005) were generated for weight gain, FCR, relative fat pad weights,

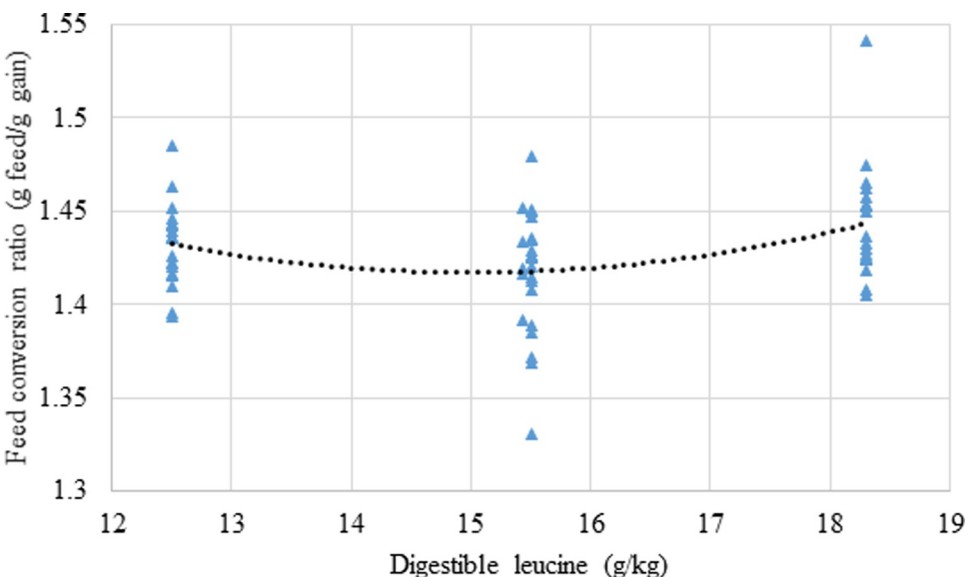

**Fig 2. Quadratic relationship (P = 0.014; r = 0.358) between dietary leucine concentrations and FCR from 7 to 28 days post-hatch where: FCR$_{(g/g, 7-28d)}$ = 1.956 – 0.0719 × digestible leucine$_{(g/kg)}$ + 0.0024 × digestible leucine$_{(g/kg)}^{2}$.** A dietary leucine concentration of 14.99 g/kg corresponds to the minimum FCR of 1.418 (g/g).

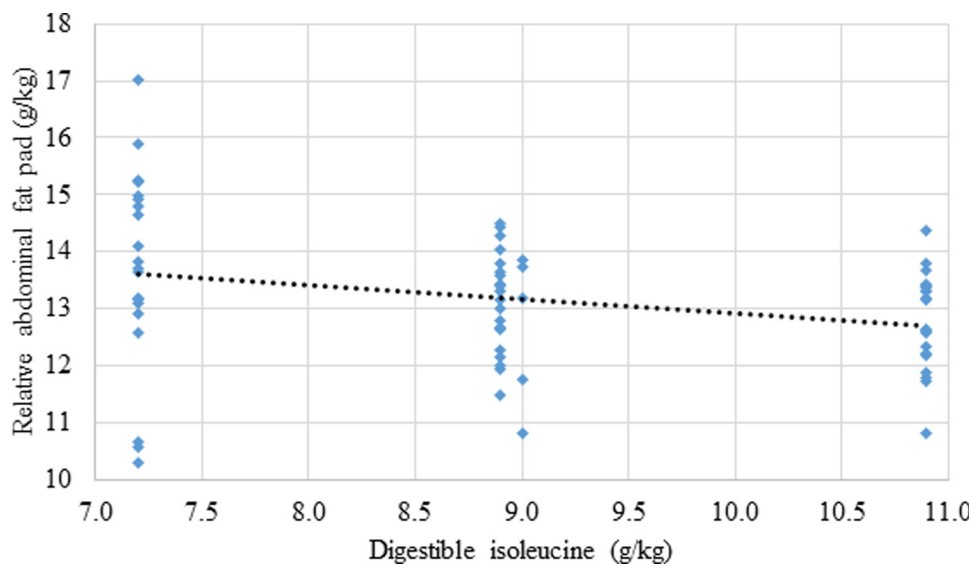

**Fig 3. Linear relationship (P = 0.022; r = -0.284) between dietary isoleucine concentrations and relative abdominal fat-pad weights where: Relative fat pad weight$_{(g/kg)}$ = 15.42–0.251 × digestible isoleucine$_{(g/kg)}$.**

AME, N retention and AME$_n$, shown in Tables 8 and 9. Surface response curves were generated for growth performance and nutrient utilisation and, with the exception of linear relationships for relative abdominal fat pad weights, the remaining parameters were quadratic. Weight gain was influenced by second order polynomial regression and interactions between BCAA as illustrated in Fig 4. Interestingly, FCR was influenced by the same second and third order polynomial regression variables, producing a negatively correlated and mirror image response surface to weight gain as shown in Fig 5. Despite elevated dietary BCAA, neither maximum weight gain nor minimum FCR could be determined by RSM polynomial regression since these were outside the treatment limits. Interestingly, whilst a minimum FCR could be determined by quadratic response to digestible leucine, including the effects of valine and isoleucine by RSM suggests the true minimum FCR may be lower than 1.418. Relative fat pad weights were influenced by first order valine and isoleucine levels and their interaction. The largest relative fat pad weights were accompanied by the lowest dietary valine and isoleucine levels shown in Fig 6. Furthermore, increasing dietary valine and isoleucine in tandem depressed relative abdominal fat-pad weights and the lightest weight of 12.34 g/kg was generated by 10.9 g/kg isoleucine coupled with 12.5 g/kg valine. However, RSM polynomial regression was unable to predict the absolute minimum nor the maximum relative fat pad weights since these were beyond the levels of valine and isoleucine used in the treatments. Similarly, RSM polynomial

**Table 8. Polynomial fitted model (P < 0.05) for growth performance from 7 to 28 days post-hatch and nutrient utilisation from 24 to 27 days post-hatch.**

| Parameter | Response mean | R-square | Regression analysis probability | | |
|---|---|---|---|---|---|
| | | | Quadratic | Total Model | Lack of fit |
| **Weight gain (g/bird)** | 1427 | 0.19 | 0.003 | 0.021 | 0.948 |
| **FCR (g/g)** | 1.430 | 0.24 | < 0.001 | 0.010 | 0.888 |
| **Relative fat pad weight (g/kg)** | 13.17 | 0.19 | 0.339 | 0.011 | 0.817 |
| **AME (MJ/kg DM)** | 13.33 | 0.29 | 0.006 | < 0.001 | 0.197 |
| **N retention (%)** | 74.07 | 0.28 | 0.001 | 0.003 | 0.013 |
| **AMEn (MJ/kg DM)** | 12.32 | 0.46 | 0.002 | < 0.001 | 0.008 |

**Table 9. Polynomial fitted model (P < 0.05) equations for growth performance from 7 to 28 days post-hatch and nutrient utilisation from 24 to 27 days post-hatch.**

| | Weight Gain (g/bird) | FCR (g/g) | Relative fat pad weight (g/kg) | AME (MJ/kg DM) | N Retention (%) | $AME_n$ (MJ/kg DM) |
|---|---|---|---|---|---|---|
| **Variables[1]** | \multicolumn Regression coefficients | | | | | |
| **Intercept** | 1486.2 | 1.4124 | 31.379 | 14.481 | 171.76 | 10.173 |
| **First order** | | | | | | |
| $X_1$ | - | - | - | - | -6.1652 | 0.3171 |
| $X_2$ | - | - | -1.5253 | -0.1765 | -10.037 | - |
| $X_3$ | - | - | -1.8382 | - | - | - |
| **Second order** | | | | | | |
| $X_1^2$ | -1.4663 | $8.503 \times 10^{-4}$ | - | - | 0.2022 | -0.0101 |
| $X_2^2$ | - | - | - | - | 0.4779 | $-8.200 \times 10^{-3}$ |
| $X_3^2$ | 4.1887 | $-2.470 \times 10^{-3}$ | - | -0.0145 | - | -0.0154 |
| **Interactions** | | | | | | |
| $X_1X_2$ | 3.9464 | $-2.314 \times 10^{-3}$ | - | - | - | - |
| $X_1X_3$ | - | - | - | - | - | - |
| $X_2X_3$ | -7.3051 | 0.0041 | 0.1518 | 0.0202 | - | 0.0201 |

[1] $X_1$: digestible leucine; $X_2$: digestible valine; $X_3$: digestible isoleucine.

regression of nutrient utilisation predicted the highest observed AME (range from 13.07 to 13.49 MJ/kg) at the lowest combination of dietary valine and isoleucine but also the lowest AME at lowest valine and highest isoleucine levels tested, but was unable to predict the absolute maximum or minimum beyond the tested levels (Fig 6). N retention influenced N-corrected AME, where maximum $AME_n$ (12.42 MJ/kg) was estimated at 15.74 g/kg dietary leucine (first and second order variable) but was influenced by second order and cross-product valine and isoleucine (Fig 7). There were no significant parameter effects for the ratio of ME to GE. In contrast, N retention (range 71.97 to 76.47%) was influenced by first and second order polynomial regression analysis of dietary leucine and valine and the lowest N retention of 72.06% was estimated at a combination of 15.25 and 10.50 g/kg of leucine and valine respectively, shown in Fig 8.

Tables 10 and 11 show the effect of dietary treatments on apparent ileal digestibility of nine notionally essential and seven non-essential amino acids, respectively. The mean digestibility coefficients for the BCAA were effectively identical with 0.792 isoleucine, 0.795 leucine and 0.793 valine. Isoleucine digestibility coefficients ranged from 0.756 to 0.836 (10.6%), leucine from 0.732 to 0.835 (14.1%) and valine from 0.752 to 0.826 (9.84%) where the minima to maxima percentage differences are shown in parentheses. The amplitude of the differences is quite considerable; however, there were positive quadratic relationships between digestibility

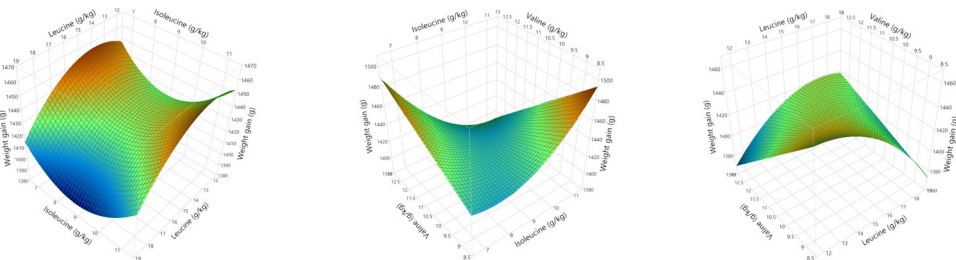

**Fig 4. Response surfaces showing the relationship between weight gain and dietary digestible branched chain amino acids in male broilers from 7 to 28 days post-hatch.**

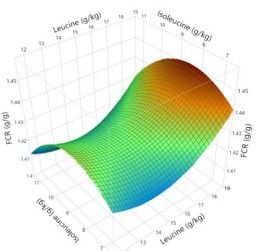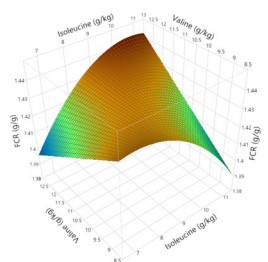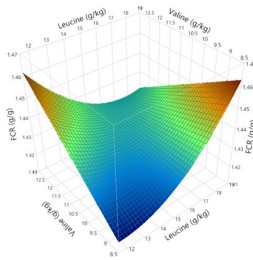

**Fig 5. Response surfaces showing the relationship between feed conversion ratio (FCR) and dietary digestible branched chain amino acids in male broilers from 7 to 28 days post-hatch.**

coefficients of leucine and isoleucine (r = 0.696; P < 0.001), leucine and valine (r = 0.692; P < 0.001), and isoleucine and valine (r = 0.713; P< 0.001). Polynomial fitted models applying RSM were significant for first and second order variables of leucine and valine with the exception of isoleucine, which included a first order isoleucine variable (Table 12). Three essential amino acids (arginine, lysine and methionine) were notable exceptions where all variables were not significant and were therefore excluded from the fitted model in Table 13. Interestingly, with the exception of isoleucine, the maximum digestibility coefficients were estimated for combined levels of valine and leucine for 12 amino acids (Table 14) and the mean response curve is illustrated in Fig 9. For isoleucine, the fitted polynomial model was unable to predict the maximum digestibility coefficient since this was beyond the level of the tested amounts. However, there was a linear response to increasing isoleucine digestibility coefficients with increasing dietary isoleucine content independent of dietary valine or isoleucine (r = 0.560; P < 0.001) shown in Fig 10.

The effect of selected dietary treatments on systemic plasma concentrations of 20 amino acids are shown in Table 15. Systemic plasma valine concentration increased linearly (P < 0.001; r = 0.992) in response to increasing dietary valine concentration. However, there was no dietary leucine response (P = 0.295) on systemic plasma leucine concentration. Interestingly, systemic plasma isoleucine peaked at the highest (18.3 g/kg) dietary leucine level in tandem with moderate (10.2 g/kg) dietary valine levels but declined (P < 0.001) at the lowest (12.5 g/kg) dietary leucine level and the same moderate dietary valine level. No further treatment effects on free amino acid systemic plasma concentrations were observed with the

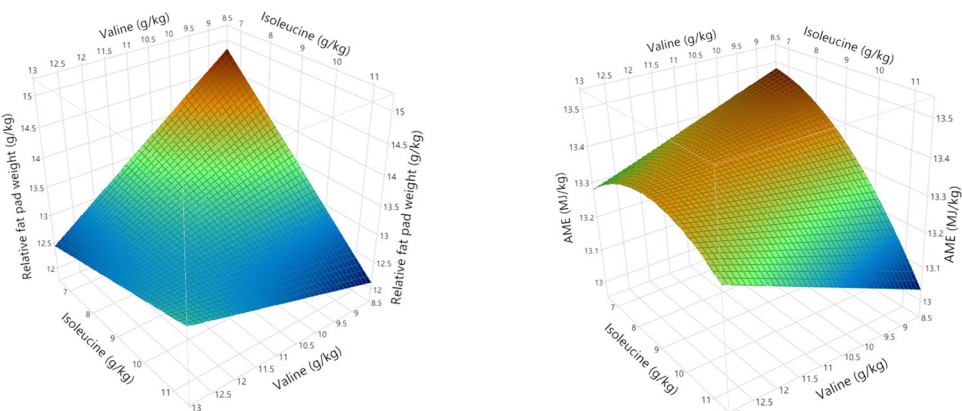

**Fig 6. Response surfaces showing the relationship between dietary digestible valine and isoleucine on relative fat pad weights (28 days post-hatch) and apparent metabolisable energy (AME) from 24 to 27 days post-hatch.**

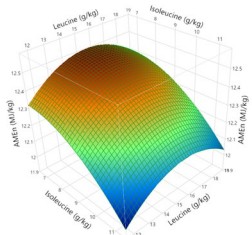 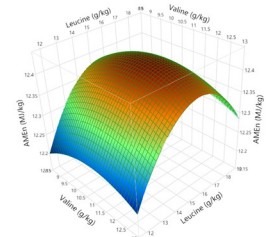 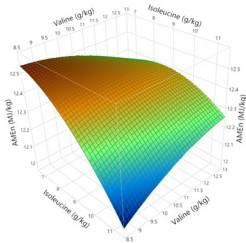

**Fig 7. Response surfaces showing the relationship between nitrogen corrected apparent metabolisable energy (AME$_n$) and dietary digestible branched chain amino acids in male broilers from 24 to 28 days post-hatch where the maximum AMEn (12.42 MJ/kg) could be estimated at 15.74 g/kg digestible leucine.**

exception of elevated (P = 0.001) phenylalanine in treatment 10 and elevated (P = 0.004) tyrosine at both the highest and lowest (18.3 and 12.5 g/kg respectively) concentrations of dietary leucine with either low (8.9 g/kg) or intermediate (10.2 g/kg) dietary valine and intermediate (8.9 g/kg) or elevated (10.9 g/kg) isoleucine.

Tables 16 and 17 show the effect of dietary treatments on apparent ileal disappearance rates of essential and non-essential amino acids, respectively. Significant differences were observed for all essential amino acids, except methionine, and for all non-essential amino acids. There were non-significant linear relationships between disappearance rates of leucine and isoleucine (r = 0.145; P = 0.248), leucine and valine (r = 0.204; P = 0.103), and isoleucine and valine (r = 0.085; P = 0.449).

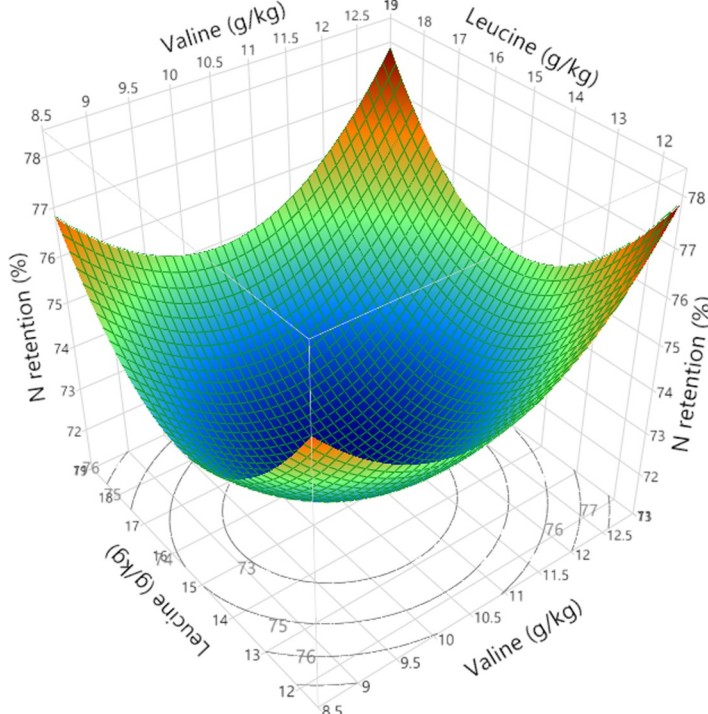

**Fig 8. Response surface showing the relationship between dietary digestible leucine (Leu) and valine (Val) on nitrogen (N) retention from 24 to 27 days post-hatch where the lowest N retention was estimated at 15.25 and 10.50 g/kg Leu and Val respectively.**

**Table 10. Effects of dietary treatments on apparent ileal digestibility coefficients of essential amino acids at 28 days post-hatch.**

| Treatment | Arginine | Histidine | Isoleucine | Leucine | Lysine | Methionine | Phenylalanine | Threonine | Valine |
|---|---|---|---|---|---|---|---|---|---|
| 1 | 0.806 | 0.720 | 0.805 | 0.773 | 0.772 | 0.873 | 0.694 | 0.682 | 0.802 |
| 2 | 0.838 | 0.764 | 0.836 | 0.816 | 0.814 | 0.890 | 0.759 | 0.735 | 0.786 |
| 3 | 0.834 | 0.753 | 0.771 | 0.803 | 0.811 | 0.886 | 0.744 | 0.725 | 0.826 |
| 4 | 0.827 | 0.751 | 0.767 | 0.807 | 0.800 | 0.887 | 0.749 | 0.730 | 0.778 |
| 5 | 0.812 | 0.729 | 0.774 | 0.809 | 0.779 | 0.872 | 0.710 | 0.697 | 0.808 |
| 6 | 0.799 | 0.705 | 0.764 | 0.732 | 0.761 | 0.863 | 0.690 | 0.671 | 0.790 |
| 7 | 0.810 | 0.727 | 0.778 | 0.810 | 0.780 | 0.876 | 0.714 | 0.698 | 0.753 |
| 8 | 0.816 | 0.730 | 0.779 | 0.751 | 0.787 | 0.879 | 0.724 | 0.703 | 0.752 |
| 9 | 0.838 | 0.762 | 0.836 | 0.835 | 0.809 | 0.888 | 0.749 | 0.729 | 0.803 |
| 10 | 0.841 | 0.765 | 0.831 | 0.782 | 0.811 | 0.891 | 0.754 | 0.738 | 0.800 |
| 11 | 0.843 | 0.761 | 0.776 | 0.832 | 0.813 | 0.893 | 0.752 | 0.739 | 0.804 |
| 12 | 0.815 | 0.741 | 0.756 | 0.759 | 0.793 | 0.876 | 0.731 | 0.713 | 0.784 |
| 13 | 0.850 | 0.783 | 0.818 | 0.827 | 0.820 | 0.900 | 0.771 | 0.765 | 0.818 |
| SEM | 0.0166 | 0.0180 | 0.0159 | 0.0136 | 0.0205 | 0.0112 | 0.0185 | 0.0188 | 0.0155 |

## Discussion

The overall growth performance of the off-sex males in the present study was marginally inferior to the Ross 308 fast feathering broiler male performance objectives (Aviagen, 2019) in terms of weight gain (1427 versus 1444 g/bird), feed intake (2040 versus 2033 g/bird) and FCR (1.430 versus 1.408). These data suggested that elevated BCAA ratios in reduced crude protein diets did not disadvantage male broiler performance. Instructively, Kidd et al. [31] investigated lower levels of BCAA in a Box-Behnken design using isoleucine levels of 58, 66 and 74 g/kg, leucine levels of 65, 75 and 85 g/kg and valine levels of 65, 75 and 85 g/kg, respectively. In agreement with the current study, these authors observed that overall male broiler performance was not affected by dietary BCAA levels but were able to demonstrate a quadratic response in abdominal fat deposition and a trend towards improved carcass yield at the lowest

**Table 11. Effects of dietary treatments on apparent ileal digestibility coefficients of non-essential amino acids at 28 days post-hatch.**

| Treatment | Alanine | Aspartic acid | Cysteine | Glutamic acid | Glycine | Proline | Serine |
|---|---|---|---|---|---|---|---|
| 1 | 0.692 | 0.645 | 0.592 | 0.749 | 0.779 | 0.680 | 0.598 |
| 2 | 0.755 | 0.714 | 0.650 | 0.799 | 0.808 | 0.743 | 0.681 |
| 3 | 0.736 | 0.703 | 0.641 | 0.787 | 0.807 | 0.733 | 0.667 |
| 4 | 0.746 | 0.706 | 0.664 | 0.796 | 0.806 | 0.750 | 0.682 |
| 5 | 0.706 | 0.665 | 0.607 | 0.766 | 0.788 | 0.695 | 0.630 |
| 6 | 0.695 | 0.646 | 0.576 | 0.750 | 0.763 | 0.677 | 0.611 |
| 7 | 0.715 | 0.665 | 0.620 | 0.767 | 0.784 | 0.710 | 0.635 |
| 8 | 0.721 | 0.674 | 0.613 | 0.770 | 0.784 | 0.700 | 0.648 |
| 9 | 0.748 | 0.706 | 0.629 | 0.791 | 0.805 | 0.725 | 0.673 |
| 10 | 0.750 | 0.720 | 0.657 | 0.801 | 0.808 | 0.732 | 0.704 |
| 11 | 0.747 | 0.726 | 0.666 | 0.804 | 0.811 | 0.736 | 0.698 |
| 12 | 0.731 | 0.695 | 0.635 | 0.780 | 0.793 | 0.713 | 0.664 |
| 13 | 0.777 | 0.737 | 0.702[d] | 0.815 | 0.822 | 0.765 | 0.732 |
| SEM | 0.0184 | 0.0216 | 0.0221 | 0.0156 | 0.0138 | 0.0160 | 0.0229 |

[abcdef] Means within columns not sharing a common superscript are significantly different at the 5% level of probability.

Table 12. Polynomial fitted model (P < 0.05) for apparent digestibility coefficients of amino acids at 28 days post-hatch.

| Amino acid | Response mean digestibility coefficients | R-square | Regression analysis probability | | |
|---|---|---|---|---|---|
| | | | Quadratic | Total Model | Lack of fit |
| Arginine | 0.825 | 0.18 | 0.927 | 0.259 | 0.894 |
| Histidine | 0.745 | 0.21 | 0.021 | 0.006 | 0.949 |
| Isoleucine | 0.792 | 0.42 | 0.047 | < 0.001 | 0.945 |
| Leucine | 0.795 | 0.53 | 0.004 | < 0.001 | 0.901 |
| Lysine | 0.796 | 0.16 | 0.077 | 0.337 | 0.917 |
| Methionine | 0.883 | 0.13 | 0.079 | 0.494 | 0.955 |
| Phenylalanine | 0.734 | 0.23 | 0.004 | 0.003 | 0.900 |
| Threonine | 0.717 | 0.25 | 0.012 | 0.002 | 0.904 |
| Valine | 0.793 | 0.29 | 0.026 | < 0.001 | 0.921 |
| Alanine | 0.732 | 0.24 | 0.019 | 0.002 | 0.920 |
| Aspartic acid | 0.693 | 0.24 | 0.030 | 0.002 | 0.911 |
| Cysteine | 0.635 | 0.28 | 0.004 | < 0.001 | 0.834 |
| Glutamic acid | 0.783 | 0.22 | 0.035 | 0.004 | 0.895 |
| Glycine | 0.797 | 0.19 | 0.018 | 0.012 | 0.907 |
| Proline | 0.719 | 0.30 | 0.006 | < 0.001 | 0.852 |
| Serine | 0.663 | 0.29 | 0.023 | < 0.001 | 0.798 |
| Mean values | 0.756 | 0.21 | 0.018 | 0.005 | 0.952 |

levels of dietary isoleucine and leucine. In contrast, maximum weight gains (and minimum FCR) in the current study, were achieved at the lowest levels of dietary leucine and valine. However, at the lowest level of valine and isoleucine, weights were depressed and FCR increased, which were similar for the highest levels of these two BCAA. The highest weights and lowest FCR trend were also observed at dietary BCAA levels of 15.4 (1.40), 10.7 (0.97) and 9.05 g/kg (0.82) for leucine, valine and isoleucine respectively (ratios to digestible lysine in parenthesis) suggesting that male broiler growth performance is enhanced in reduced crude protein diets, when BCAA is in excess of the primary breeder recommendations. Broiler growth response to increasing dietary valine was dependant on both the dietary leucine and isoleucine levels suggesting interaction between these BCAA. At the highest dietary leucine content, increasing valine decreased FCR and increased weight gain but, the opposite occurred at the lowest leucine treatment. Conversely, increasing dietary valine at the highest isoleucine increased FCR and decreased weight gain whilst FCR declined and weight gain increased when valine was increased at the lowest isoleucine treatment level. From these data, it is evident that responses in broiler growth are influenced by the BCAA and that both the ratios and dietary levels should be considered in reduced protein diets.

Maynard et al. [32] recently evaluated BCAA via Box-Behnken RSM using similar levels to Kidd et al. [31] and observed no influence of BCAA on broiler growth performance but demonstrated a trend towards increasing FCR with increasing dietary leucine levels which is in agreement with the current study. Relative fat pad weights declined with increasing dietary isoleucine concentration but only at the lowest dietary valine concentration. At the highest dietary valine, increasing isoleucine had a small effect on decreased relative fat pad weight. The higher than usual isoleucine and leucine levels in this study may partially explain this observation. This is in agreement with Maynard et al. [32] who observed a trend for a quadratic decline in relative abdominal fat pad with increasing dietary valine concentrations. Furthermore, in agreement with Kidd et al. [31] increased dietary isoleucine alone in the present study, reduced relative abdominal fat pad weight in male broilers. Subsequently, Greenhalgh

**Table 13. Polynomial fitted model ($P < 0.05$) equations for apparent digestibility coefficients of amino acids at 28 days post-hatch.**

| Variables[1] | Histidine | Isoleucine | Leucine | Phenyl alanine | Threonine | Valine | Alanine | Aspartic acid | Cysteine | Glutamic acid | Glycine | Proline | Serine |
|---|---|---|---|---|---|---|---|---|---|---|---|---|---|
| | | | | | | | Regression coefficients | | | | | | |
| **Intercept** | -1.1368 | -0.6790 | -0.9062 | -1.4200 | -1.4082 | -0.8442 | -1.1782 | -1.6793 | -2.0592 | -0.7832 | -0.5516 | -1.1779 | -2.1903 |
| **First order** | | | | | | | | | | | | | |
| $X_1$ | 0.0914 | 0.0688 | 0.0972 | 0.0974 | 0.1044 | 0.0753 | 0.0950 | 0.1037 | 0.1442 | 0.0731 | 0.0726 | 0.1256 | 0.1169 |
| $X_2$ | 0.2293 | 0.1547 | 0.1722 | 0.2233 | 0.2585 | 0.1900 | 0.2337 | 0.3081 | 0.3124 | 0.1973 | 0.1533 | 0.1870 | 0.3846 |
| $X_3$ | - | 0.0161 | - | - | - | - | - | - | - | - | - | - | - |
| **Second order** | | | | | | | | | | | | | |
| $X_1^2$ | $-2.916 \times 10^{-3}$ | $-2.185 \times 10^{-3}$ | $-2.792 \times 10^{-3}$ | $-3.124 \times 10^{-3}$ | $-3.334 \times 10^{-3}$ | $-2.386 \times 10^{-3}$ | $-3.058 \times 10^{-3}$ | $-3.329 \times 10^{-3}$ | $-4.626 \times 10^{-3}$ | $-2.337 \times 10^{-3}$ | $-2.300 \times 10^{-3}$ | $-4.029 \times 10^{-3}$ | $-3.780 \times 10^{-3}$ |
| $X_2^2$ | -0.0109 | $-7.374 \times 10^{-3}$ | $-8.260 \times 10^{-3}$ | -0.0108 | -0.0124 | $-8.373 \times 10^{-3}$ | $-1.127 \times 10^{-3}$ | -0.0147 | -0.0150 | $-9.480 \times 10^{-3}$ | $-7.313 \times 10^{-3}$ | $-9.120 \times 10^{-3}$ | -0.0184 |
| $X_3^2$ | - | - | - | - | - | - | - | - | - | - | - | - | - |

[1] $X_1$: Digestible leucine; $X_2$: Digestible valine; $X_3$: Digestible isoleucine.

Table 14. Estimated maximum digestibility coefficients by dietary leucine and valine levels at 28 days post-hatch.

| Parameter | Histidine | Leucine | Phenyl alanine | Threonine | Valine | Alanine | Aspartic acid | Cysteine | Glutamic acid | Glycine | Proline | Serine | Mean |
|---|---|---|---|---|---|---|---|---|---|---|---|---|---|
| Digestibility coefficient | 0.783 | 0.837 | 0.773 | 0.760 | 0.828 | 0.772 | 0.740 | 0.690 | 0.814 | 0.824 | 0.760 | 0.721 | 0.793 |
| Leucine (g/kg) | 15.67 | 17.41 | 15.59 | 15.66 | 15.79 | 15.54 | 15.57 | 15.59 | 15.64 | 15.77 | 15.59 | 15.46 | 15.74 |
| Valine (g/kg) | 10.50 | 10.42 | 10.35 | 10.45 | 11.34 | 10.37 | 10.46 | 10.40 | 10.40 | 10.48 | 10.25 | 10.44 | 10.47 |

et al. [25] reported a minimum relative fat pad weight of 52.2 g/kg generated by a combined isoleucine and valine concentration of 20.71 g/kg in wheat-based diets. Indeed, dietary valine was reported to decrease fatty acid synthesis without stimulating lipid degradation [1,25]. Isoleucine and leucine had similar role in lipid metabolism, with leucine decreasing blood triglyceride concentrations [1,33].

Weight gain from 7 to 28 days post-hatch tended to be positively correlated with $AME_n$ (P = 0.055; r = 0.239) but was not correlated to AME (P = 0.875; r = 0.020) nor N retention

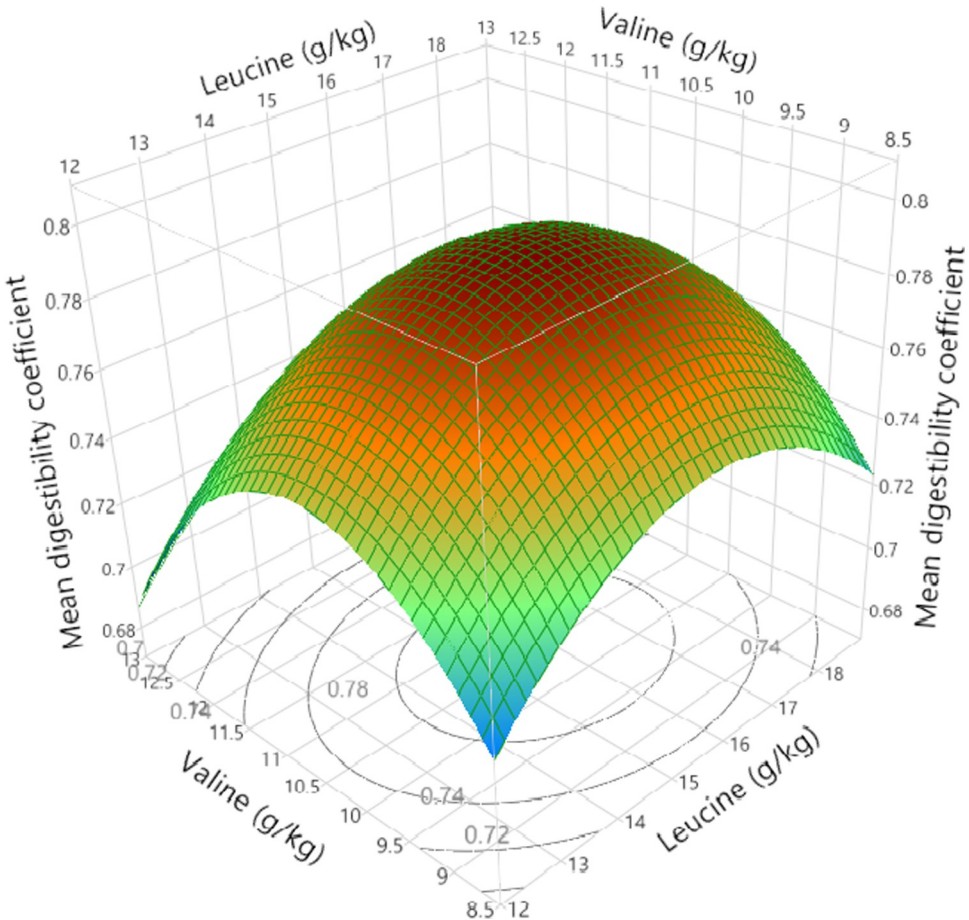

Fig 9. Polynomial fitted model surface response influence (P = 0.011; r = 0.313) of dietary digestible (dig.) leucine (leu) and dig. valine (Val) on mean amino acid (AA) digestibility coefficients where: Mean AA digestibility$_{(coefficient)}$ = 0.0906 × dig. Leu$_{(g/kg)}$ + 0.2153 × dig. Val$_{(g/kg)}$ − 2.879 × 10$^{-3}$ × dig. Leu$_{(g/kg)}^2$ − 0.0103 × dig. Val$_{(g/kg)}^2$ − 1.0478. The maximum mean digestibility coefficient of 0.793 was estimated at a combination of 15.74 and 10.47 g/kg of digestible leucine and valine respectively.

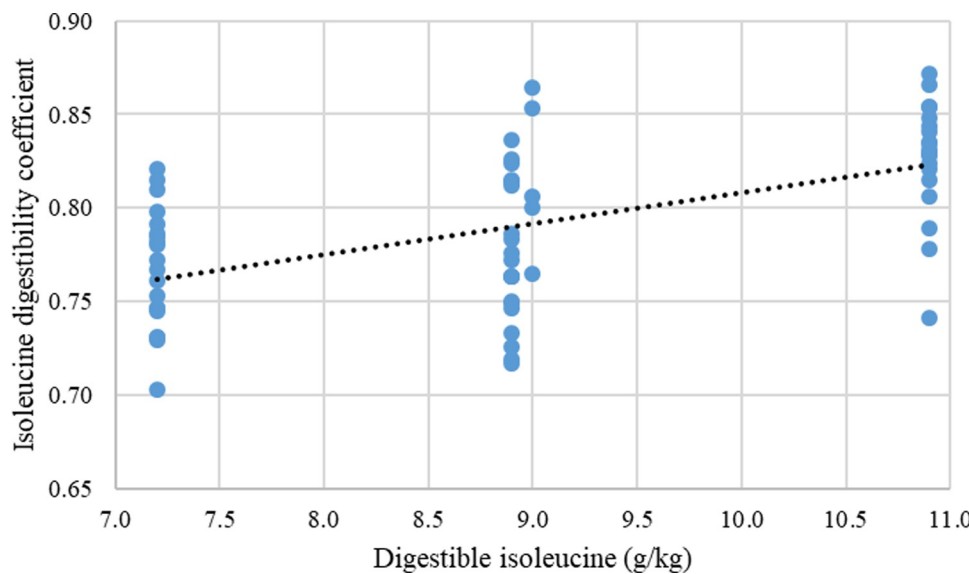

**Fig 10. Linear relationship (P < 0.001; r = 0.560) of dietary digestible isoleucine on isoleucine digestible coefficient where: Digestible isoleucine$_{(coefficient)}$ = 0.6443 + 0.0164 × digestible isoleucine$_{(g/kg)}$.**

(P = 0.953; r = 0.007). Similarly, with the exception of the influence of AME on relative fat pad weights (P = 0.027; r = 0.275) feed intake, resultant FCR and fat pad weights were not correlated with any other parameters of nutrient utilisation (P range from 0.155 to 0.953 and r from

**Table 15. Effects of selected dietary treatments on free amino acid systemic plasma concentrations (µg/mL) at 27 days post-hatch.**

| Dietary treatment | Arginine | Histidine | Isoleucine | Leucine | Lysine | Methionine | Phenylalanine |
|---|---|---|---|---|---|---|---|
| 5E | 54.0 | 3.8 | 11.4[b] | 32.8 | 47.8 | 23.2 | 11.2[a] |
| 7G | 48.8 | 3.5 | 11.3[b] | 32.8 | 45.0 | 22.3 | 12.5[a] |
| 9I | 52.8 | 3.8 | 18.9[c] | 35.8 | 51.0 | 24.8 | 11.8[a] |
| 10J | 53.6 | 3.8 | 7.6[a] | 34.8 | 42.8 | 20.4 | 14.8[b] |
| SEM | 2.943 | 0.230 | 0.590 | 1.254 | 4.268 | 1.244 | 0.525 |
| **Dietary treatment** | **Threonine** | **Tryptophan** | **Valine** | **Alanine** | **Asparagine** | **Aspartic acid** | **Cysteine** |
| 5E | 130.4 | 5.8 | 47.0[c] | 81.6 | 32.2 | 7.6 | 11.0 |
| 7G | 110.5 | 6.0 | 21.8[a] | 81.3 | 30.8 | 8.5 | 11.8 |
| 9I | 131.2 | 5.5 | 33.3[b] | 84.8 | 32.5 | 8.3 | 11.3 |
| 10J | 126.2 | 6.4 | 32.2[b] | 75.8 | 27.4 | 10.1 | 12.0 |
| SEM | 9.073 | 0.282 | 1.438 | 4.793 | 2.891 | 0.766 | 0.527 |
| **Dietary treatment** | **Glutamine** | **Glutamic acid** | **Glycine** | **Proline** | **Serine** | **Tyrosine** | **Total** |
| 5E | 214.6 | 18.6 | 120.3 | 44.6 | 67.4 | 17.4[a] | 982.8 |
| 7G | 226.0 | 20.0 | 112.0 | 43.0 | 68.0 | 21.0[ab] | 926.3 |
| 9I | 224.5 | 20.3 | 115.0 | 45.3 | 65.0 | 18.0[a] | 994.0 |
| 10J | 197.4 | 19.2 | 108.8 | 43.6 | 70.8 | 24.0[b] | 931.2 |
| SEM | 12.863 | 0.932 | 4.784 | 1.848 | 2.357 | 1.190 | 32.160 |

[abc] Means within columns not sharing a common superscript are significantly different at the 5% level of probability.

**Table 16. Effects of dietary treatments on apparent disappearance rates (g/bird/day) of essential amino acids at 28 days post-hatch.**

| Treatment | Arginine | Histidine | Isoleucine | Leucine | Lysine | Methionine | Phenylalanine | Threonine | Valine |
|---|---|---|---|---|---|---|---|---|---|
| 1 | 0.92[abc] | 0.28[ab] | 0.82[e] | 1.25[cd] | 0.85[abc] | 0.47 | 0.46[a] | 0.50[ab] | 0.98[de] |
| 2 | 0.95[abcd] | 0.30[bcd] | 0.86[ef] | 1.37[e] | 0.90[bcd] | 0.48 | 0.54[cd] | 0.55[bcde] | 0.75[a] |
| 3 | 0.98[cde] | 0.31[cd] | 0.61[ab] | 1.37[e] | 0.93[bcd] | 0.50 | 0.54[cd] | 0.56[cde] | 1.05[e] |
| 4 | 0.94[abcd] | 0.29[bc] | 0.59[a] | 1.34[de] | 0.88[abcd] | 0.48 | 0.54[cd] | 0.54[bcde] | 0.75[a] |
| 5 | 0.94[abcd] | 0.29[bc] | 0.68[cd] | 1.52[f] | 0.88[abcd] | 0.49 | 0.48[ab] | 0.52[abcd] | 1.01[e] |
| 6 | 0.87[a] | 0.26[a] | 0.65[abcd] | 1.02[a] | 0.79[a] | 0.45 | 0.46[a] | 0.48[a] | 0.92[cd] |
| 7 | 0.89[ab] | 0.28[ab] | 0.67[bcd] | 1.51[f] | 0.84[ab] | 0.47 | 0.49[abc] | 0.51[abc] | 0.71[a] |
| 8 | 0.95[abcd] | 0.29[bc] | 0.71[d] | 1.11[ab] | 0.89[bcd] | 0.49 | 0.54[cd] | 0.54[bcde] | 0.73[a] |
| 9 | 0.97[bcde] | 0.30[bcd] | 0.86[ef] | 1.58[fg] | 0.90[bcd] | 0.48 | 0.51[abcd] | 0.55[bcde] | 0.85[bc] |
| 10 | 1.02[d] | 0.32[d] | 0.89[f] | 1.18[bc] | 0.95[d] | 0.50 | 0.56[d] | 0.58[e] | 0.88[bc] |
| 11 | 1.03[e] | 0.31[cd] | 0.62[abc] | 1.63[g] | 0.95[d] | 0.50 | 0.55[d] | 0.59[e] | 0.90[bc] |
| 12 | 0.96[bcde] | 0.30[bcd] | 0.61[ab] | 1.13[b] | 0.94[cd] | 0.49 | 0.55[d] | 0.57[cde] | 0.86[bc] |
| 13 | 0.95[abcde] | 0.30[bcd] | 0.70[d] | 1.35[de] | 0.86[abcd] | 0.46 | 0.52[bcd] | 0.57[cde] | 0.84[b] |
| SEM | 0.0291 | 0.0102 | 0.0216 | 0.0385 | 0.0321 | 0.0321 | 0.0184 | 0.0196 | 0.0261 |

[abcdefg] Means within columns not sharing a common superscript are significantly different at the 5% level of probability.

0.002 to 0.178). However, there was a positive linear correlation ($P < 0.001$; $r = 0.424$) between percent N retention and observed dietary AME from 24 to 27 days post-hatch described by the following equation where,

$$\text{N retention}_{(\%)} = 6.986 + \text{AME}_{(MJ/kg)} \times 5.035$$

Importantly, nutrient utilisation expressed as AME, $\text{AME}_n$ and N retention were poor predictors of broiler performance but AME was a reliable predictor of N retention.

**Table 17. Effects of dietary treatments on apparent disappearance rates (g/bird/day) of non-essential amino acids at 28 days post-hatch.**

| Treatment | Alanine | Aspartic acid | Cysteine | Glutamic acid | Glycine | Proline | Serine |
|---|---|---|---|---|---|---|---|
| 1 | 0.52[a] | 0.81[a] | 0.15[ab] | 1.83[a] | 0.84[bc] | 0.61[ab] | 0.37[a] |
| 2 | 0.60[cd] | 0.95[cde] | 0.16[abc] | 2.06[bcde] | 0.85[bc] | 0.70[de] | 0.45[cde] |
| 3 | 0.59[bcd] | 0.96[de] | 0.17[bc] | 2.06[bcde] | 0.90[c] | 0.73[e] | 0.44[bcd] |
| 4 | 0.59[bcd] | 0.96[de] | 0.17[bc] | 2.08[de] | 0.85[bc] | 0.72[e] | 0.45[cde] |
| 5 | 0.54[ab] | 0.85[abc] | 0.16[abc] | 1.90[ab] | 0.86[bc] | 0.63[abc] | 0.40[abc] |
| 6 | 0.54[ab] | 0.83[ab] | 0.14[a] | 1.85[a] | 0.76[a] | 0.60[a] | 0.39[ab] |
| 7 | 0.55[abc] | 0.85[abc] | 0.16[abc] | 1.92[abc] | 0.82[b] | 0.65[abc] | 0.40[abc] |
| 8 | 0.59[bcd] | 0.94[cde] | 0.16[abc] | 2.07[bcde] | 0.84[bc] | 0.68[cde] | 0.44[bcd] |
| 9 | 0.59[bcd] | 0.92[bcd] | 0.16[abc] | 2.00[abcd] | 0.86[bc] | 0.66[abcd] | 0.44[bcd] |
| 10 | 0.62[d] | 1.03[e] | 0.18[c] | 2.19[e] | 0.88[bc] | 0.70[de] | 0.52[f] |
| 11 | 0.61[d] | 1.04[e] | 0.18[c] | 2.20[e] | 0.90[c] | 0.69[cde] | 0.50[ef] |
| 12 | 0.61[d] | 1.02[de] | 0.18[c] | 2.17[de] | 0.88[bc] | 0.67[bcde] | 0.49[def] |
| 13 | 0.61[d] | 0.96[de] | 0.18[c] | 2.05[bcde] | 0.83[bc] | 0.70[de] | 0.50[ef] |
| SEM | 0.0208 | 0.0376 | 0.0072 | 0.0622 | 0.0250 | 0.0218 | 0.0196 |

[abcdef] Means within columns not sharing a common superscript are significantly different at the 5% level of probability.

The systemic plasma concentration of the aromatic amino acids (AAA) phenylalanine and tyrosine were elevated at the lowest systemic plasma concentrations of isoleucine. Conversely, the lowest systemic plasma levels of AAA were observed at the highest systemic plasma concentration of valine whilst leucine remained unaffected. The heterodimeric SLC7A5 (or LAT1) transporter is responsible for the uptake of both BCAA and AAA across the blood brain barrier and mediates a pH and $Na^+$ independent antiport of amino acids [34] suggesting that competition for transporters between BCAA and AAA is occurring in broilers. Broiler chickens contain almost double the level of blood glucose compared with mammals [35] and the impairment of insulin sensitivity prior to observations of hyperglycaemia are related to altered BCAA and AAA metabolism [36]. This is relevant in tangibly reduced crude protein diets since starch levels typically exceed 420 g/kg, whilst normal maize/soy-based or wheat/soy-based broiler grower diets are lower in starch content, in the order of 326 and 306 g/kg, respectively [37].

Thus, a distinct possibility exists that glucose and amino acids may compete for intestinal uptake via $Na^+$-dependent transporters, influencing nutrient utilisation as evidenced by the range of AME, $AME_n$ and N retention observed in the current study. Importantly, the highest and the lowest AME were observed with the lowest dietary valine, intermediate or high leucine and the lowest and highest dietary isoleucine respectively.

The increase in apparent ileal digestibility of the individual BCAA with increasing dietary concentration occurred irrespective of levels of the remaining two BCAA. A possible explanation for this may be the N dynamics of reduced protein diets. Diets were formulated to be iso-nitrogenous and l-valine, l-isoleucine and l-leucine were formulated applying crude protein ($N_{(g/kg)} \times 6.25$) matrix values of 732, 640 and 657 g/kg respectively. It is thus possible that antagonism, measured by apparent digestibility coefficients, is negated when dietary nitrogen is maintained in tangibly reduced crude protein diets. The underlying mechanism for the influence of BCAA levels on the apparent digestibility of the non-essential aspartic acid, cysteine, proline and serine is unclear and future elucidation is warranted. One possible explanation is that increasing dietary BCAA inclusions may upregulate intestinal transport systems influencing these non-essential amino acids due to over-lapping specificities. Nevertheless, the highest apparent digestibility of all four of these amino acids occurred at the centre-point treatment (number 13) suggesting this treatment provided the ideal ratios between BCAA in the current study. Conversely, the lowest apparent digestibility of these amino acids were observed at treatments one or six where dietary valine was at the highest level. Furthermore, maximum digestibility coefficients for the essential amino acids histidine, leucine, phenylalanine and valine were estimated from RSM polynomial regression indicating that ideal levels of leucine and valine were within the treatment range.

Whilst the disappearance rates of the individual BCAA occurred at treatments containing their highest dietary inclusion rate, this was not observed for all four treatments with the respective highest dietary BCAA inclusions. Indeed, for leucine this occurred in treatment 11 with valine at an intermediate level and isoleucine at the lowest level. For valine, this occurred in treatment three when dietary leucine was at an intermediate level and isoleucine was at the lowest level. Similarly, for isoleucine this occurred in treatment 10 where dietary valine was intermediate and leucine was at the lowest level. For all three BCAA, lowest disappearance rates were observed at treatments with the lowest level of the relevant BCAA. The lowest disappearance rate for isoleucine was accompanied by the lowest dietary valine and an intermediate level of leucine whilst the lowest disappearance rate for valine was in tandem with the lowest dietary leucine level and intermediate dietary isoleucine level. In contrast, the lowest disappearance rate for leucine was accompanied by the highest dietary valine coupled with intermediate isoleucine. Notably, excluding the BCAA, treatment six (elevated valine) depressed the

apparent disappearance rates of five essential (arginine, histidine, lysine, phenylalanine and threonine) and four non-essential (cysteine, glycine, proline and serine) amino acids. Also, the highest disappearance rates for arginine, histidine, lysine, phenylalanine, threonine, alanine, aspartic acid, cysteine, glutamic acid, glycine and serine were observed when dietary valine content was intermediate and this was independent of dietary leucine or isoleucine. Proline was the exception, having the highest apparent disappearance rate at the lowest dietary isoleucine levels.

Digestive dynamics include the rate and extent of nutrient digestion in the gastro-intestinal tract, absorption of end-products into the gut mucosa, and their entry into the systemic circulation. This process implies both static components, such as apparent ileal digestibility and kinetic aspects of rate, quantity and site of absorption of glucose and amino acids along the small intestine should be considered [38]. BCAA had a profound effect on the kinetic aspects of amino acid digestion in the current study and influenced nutrient utilisation as measured by AME, $AME_n$ and N retention. Dietary treatments were similar in maize and starch contents implying similar digestive dynamics for glucose across all treatments. There is the possibility that BCAA could be used to retard the rate of amino acid digestion in rapidly digested starch, wheat-based diets. The range in dietary supplemented non-bound amino acids was substantial and differed by 60.3% across treatments (from 26.2 to 41.6 g/kg; average, 34.1 g/kg). Despite the large difference between the maximum (treatment five) and the minimum (treatment 12) non-bound supplemented amino acid levels, with the exception of phenylalanine, there were only trends towards greater disappearance rates in the essential amino acids (excluding BCAA). However, increases in disappearance rates for the non-essential amino acids were significant for alanine, aspartic acid, glutamic acid and serine in treatment 12 compared with treatment five. Although the underlying mechanisms for these differences are not clear, it is evident that the BCAA ratios have a greater influence on digestive dynamics than dietary non-bound amino acid content *per se*.

## Conclusions

In the present study, when individual BCAA levels are in excess of requirements, antagonistic effects on broiler growth performance and nutrient utilisation were evident applying RSM, polynomial regression analysis to tangibly reduced crude protein diets. Interestingly for all parameters measured, with the exception of amino acid digestibility and N retention, minima and maxima responses were unable to be determined since these were beyond the treatment BCAA levels. Thus, BCAA levels in considerable excess of accepted requirements may elicit desired responses and the ultimate decision on what levels to use in feed should be an economic one. Also, lipid metabolism influenced by dietary BCAA levels has implications for lean meat yield which could be important economically and should be considered. Further investigation is required to better understand the dynamics between protein-bound and non-bound amino acids digestion and absorption in the context of reduced crude protein diets.

## Supporting information

**S1 File. Cage means and analyses of experimental parameters.**
(XLSX)

## Acknowledgments

The authors would like to acknowledge Ms Preethi Ramesh and her colleagues in the Evonik AMINO*Lab*® in Singapore for their analyses of amino acid concentrations in diets and

digesta. Similarly, we would like to acknowledge Dr Bernie McInerney and Dr Leon McQuade and their colleagues in the Australian Proteome Analytical Facility within Macquarie University for their analyses of free amino acid concentrations in plasma. As always, the authors also acknowledge Ms Joy Gill, Mr Wade Chen, Ms Kylie Warr and Mr Peter Bird in the Poultry Research Foundation for their invaluable technical support.

## Author Contributions

**Conceptualization:** Peter V. Chrystal, Peter H. Selle, Sonia Y. Liu.

**Data curation:** Shiva Greenhalgh, Sonia Y. Liu.

**Formal analysis:** Peter V. Chrystal, Peter H. Selle.

**Funding acquisition:** Juliano C. de Paula Dorigam.

**Investigation:** Peter V. Chrystal, Peter H. Selle, Sonia Y. Liu.

**Methodology:** Peter V. Chrystal, Peter H. Selle, Sonia Y. Liu.

**Project administration:** Juliano C. de Paula Dorigam, Sonia Y. Liu.

**Visualization:** Peter V. Chrystal, Shemil P. Macelline, Peter H. Selle.

**Writing – original draft:** Peter V. Chrystal.

**Writing – review & editing:** Peter V. Chrystal, Shiva Greenhalgh, Shemil P. Macelline, Juliano C. de Paula Dorigam, Peter H. Selle, Sonia Y. Liu.

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
