## [Decision Letter · Decision Letter 0]

14 Jun 2021

PONE-D-21-12314

A multivariate Box-Behnken assessment of elevated branched-chain amino acid concentrations in reduced crude protein diets offered to male broiler chickens

PLOS ONE

Dear Dr. Liu,

Thank you for submitting your manuscript to PLOS ONE. After careful consideration, we feel that it has merit but does not fully meet PLOS ONE’s publication criteria as it currently stands. Therefore, we invite you to submit a revised version of the manuscript that addresses the points raised during the review process.

We look forward to receiving your revised manuscript.

Kind regards,

Juan J Loor

Academic Editor

PLOS ONE

Journal Requirements:

'The study is funded by Evonik Nutrition and Care GmbH, Hanau-Wolfgang 63457, Germany'

We note that you received funding from a commercial source: Evonik Nutrition and Care GmbH

'The authors have no conflicts of interest to declare.'  

We note that one or more of the authors are employed by a commercial company: Baiada Poultry Pty Limited and Evonik Operations GmbH.

Additional Editor Comments (if provided):

Reviewers' comments:

Reviewer's Responses to Questions

**Comments to the Author**

1. Is the manuscript technically sound, and do the data support the conclusions?

Reviewer #1: No

2. Has the statistical analysis been performed appropriately and rigorously? 

Reviewer #1: No

3. Have the authors made all data underlying the findings in their manuscript fully available?

Reviewer #1: Yes

4. Is the manuscript presented in an intelligible fashion and written in standard English?

Reviewer #1: Yes

5. Review Comments to the Author

Reviewer #1: This manuscript is a potential addition to several recent papers attempting to better understand branched-chain amino acid (Val, Ile, and Leu) utilization, specifically in the context of reduced protein diets. This is a well-organized and written manuscript with minimal to no technical or grammatical errors. However, there are several major concerns regarding the design of the experiment.

1) Dietary amino acid (BCAA) levels tested: The recommended dietary concentrations of Val, Ile, and Leu for Ross 308 broiler for this age (per Aviagen specifications) are 0.87, 0.78, and 1.27%, respectively, with a Lys recommendation of 1.15%. Thus, ratios relative to Lys are 76, 68, and 110 for Val, Ile, and Leu. In this experiment (Table 1) Val:Lys ratios evaluated ranged from 89 to 125 (17 to 64% greater than recc.), Ile:Lys ratios ranged from 68 to 109 (6 to 60% greater than recc.), and Leu:Lys ratios ranged from 125 to 183 (14 to 66% greater than recc.).

Why were the lowest levels of each BCAA tested 6 to 17% higher than current recommendations? From a practical standpoint, least-cost formulation will dictate that digestible BCAA are at the minimum level specified, especially in reduced CP diets, and this specified minimum will likely be similar to current recommendations. From a scientific perspective, the conclusion of “…when individual BCAA are not deficient, they will not result in antagonism” (line 427) should be changed to “…when BCAA are in marked excess of current recommendations…” The practical or scientific merit of such a conclusion is not clear.

2) Formulation approach: The authors clearly took great care to balance the diets. However, the approach for doing so resulted in considerable changes in what should be a relatively constant “basal” component of the diet. For example, soy oil, which can have effects independent of calorie contribution, ranged from 2.77 to 11.1 g/kg (4x and additional ~75 kcal contribution from oil), and corn and SBM inclusions varied as well. Alternatively, BCAA could have been varied at the expense of glutamic acid and corn starch, likely allowing isonitrogenous, isocaloric, and balanced digestible AA with constant corn and SBM inclusions and only minor variations in soy oil.

3) Statistical analysis: The reviewer understands that the high number of treatments required in this design limits capacity for replication. Five replications are low but likely acceptable for the primary objectives of dose response evaluation. However, this replication was likely too low for making means comparisons. Further, the method for separating means should be stated (Tukey’s or other adjustment necessary), and if it was Fisher’s LSD (assumption based on LSD value reported in table), this would be inappropriate given the large number of treatments. Much of the results and conclusions are mean comparisons, however, lack of replication and use of unadjusted mean comparison test are concerning for Type 2 and Type 1 errors, respectively. This reviewer believes that interpretation should only be based on independent and interactive linear and quadratic dose responses and not individual means comparisons.

In summary, to be suitable for publication, this paper should be revised to bring forward and provide reasonable justification for use of such high levels of BCAA (5 to 60% excess of likely requirement), and data interpretation should be based only on dose responses. Given that primary ingredients outside of those tested varied across treatments, responses on dietary AME, N utilization, and AA digestibility should be deemphasized.

6. PLOS authors have the option to publish the peer review history of their article (what does this mean?). If published, this will include your full peer review and any attached files.

Reviewer #1: No

---

## [Author Response · Author response to Decision Letter 0]

29 Jul 2021

1. Is the manuscript technically sound, and do the data support the conclusions?

Reviewer #1: No

The Box-Behnken design (BBD) in nutrition research is largely based on de Leon et al., described in reference # [24] of the submitted manuscript. These authors applied a BBD investigating the responses of male broilers to lysine, threonine and total sulphur amino acids from 15 to 35 days post-hatch. In their study, treatments were replicated 5 times based on Myers and Montgomery (1995) recommendations. For this reason, we selected the 5 repeats as being sufficient for a BBD to demonstrate required responses/antagonisms with branched chain amino acids. 

2. Has the statistical analysis been performed appropriately and rigorously? 

Reviewer #1: No

Agree with the reviewer - In retrospect, a simple one-way ANOVA is not the correct way to do response surface methodology (RSM) and the statistics have been re-run as a BBD in JMP Pro version 15.2.0. The methodology describes externally studentized residuals with 95% simultaneous limits (Bonferroni). 

3. Have the authors made all data underlying the findings in their manuscript fully available?

Reviewer #1: Yes

4. Is the manuscript presented in an intelligible fashion and written in standard English?

Reviewer #1: Yes

5. Review Comments to the Author

Reviewer #1: This manuscript is a potential addition to several recent papers attempting to better understand branched-chain amino acid (Val, Ile, and Leu) utilization, specifically in the context of reduced protein diets. This is a well-organized and written manuscript with minimal to no technical or grammatical errors. However, there are several major concerns regarding the design of the experiment.

1) Dietary amino acid (BCAA) levels tested: The recommended dietary concentrations of Val, Ile, and Leu for Ross 308 broiler for this age (per Aviagen specifications) are 0.87, 0.78, and 1.27%, respectively, with a Lys recommendation of 1.15%. Thus, ratios relative to Lys are 76, 68, and 110 for Val, Ile, and Leu. In this experiment (Table 1) Val:Lys ratios evaluated ranged from 89 to 125 (17 to 64% greater than recc.), Ile:Lys ratios ranged from 68 to 109 (6 to 60% greater than recc.), and Leu:Lys ratios ranged from 125 to 183 (14 to 66% greater than recc.).

Why were the lowest levels of each BCAA tested 6 to 17% higher than current recommendations? From a practical standpoint, least-cost formulation will dictate that digestible BCAA are at the minimum level specified, especially in reduced CP diets, and this specified minimum will likely be similar to current recommendations. From a scientific perspective, the conclusion of “…when individual BCAA are not deficient, they will not result in antagonism” (line 427) should be changed to “…when BCAA are in marked excess of current recommendations…” The practical or scientific merit of such a conclusion is not clear.

Normal grower diets based on maize/soyabean meal never minimize on digestible leucine. Typically, these diets have a ratio of digestible leucine: digestible lysine greater than 1.25:1. If one uses DDGS, then the ratio widens further in standard diets. Since this trial was aimed at testing branched chain amino acid (BCAA) antagonism, the starting point was selection of a typical dietary leucine level, even though, as dietary CP declines, the minimum ratios of all BCAA:Lys tend to be approached. The trial was designed specifically to test high levels of BCAA on broiler growth performance, digestive dynamics and blood plasma profiles. Thus the initial BCAA levels were chosen to exceed the primary breeder requirements. The wide range of BCAA ratios tested should have resulted in substantial antagonisms of BCAA or responses to BCAA – if they exist in reduced protein diets. 

The conclusion was changed as suggested. 

2) Formulation approach: The authors clearly took great care to balance the diets. However, the approach for doing so resulted in considerable changes in what should be a relatively constant “basal” component of the diet. For example, soy oil, which can have effects independent of calorie contribution, ranged from 2.77 to 11.1 g/kg (4x and additional ~75 kcal contribution from oil), and corn and SBM inclusions varied as well. Alternatively, BCAA could have been varied at the expense of glutamic acid and corn starch, likely allowing isonitrogenous, isocaloric, and balanced digestible AA with constant corn and SBM inclusions and only minor variations in soy oil.

The diets were deliberately formulated to mimic commercial-type diets so corn starch was avoided. However, the AME of glutamic acid is about half that of the BCAA and maize starch about two-thirds, so one cannot get iso-energetic diets with that combination. The crude protein matrix values used for “as fed” are 595 g/kg for glutamic acid and, 732.3, 640.0 and 657.0 g/kg for Val, Ile and Leu respectively so, once again it is difficult to get iso-nitrogenous diets with an amino acid that has a lower N content than the test BCAA. Also, there is some evidence to suggest that glutamic acid itself could confound the trial. Previous work on BCAA often excludes a CP value for the test AA and then simply utilise an inert filler. In our view, this changes the fundamental balance of the diet since dietary CP would change as would AME by treatment. This would therefore confound the effects of just the 3 BCAA levels being tested. 

As far as dietary added oil is concerned, the authors note that all diets contained canola seed at 60 g/kg supplying ~ 25 g/kg oil, so the soya oil is simply added as a top-up to balance the dietary AME and ensure these diets are all iso-caloric. The total crude fat of the diets ranged from 59.0 to 67.4 g/kg (treatments 5 & 12) representing a difference of 14.2%. Once again, this would not be seen as excessive and is well within normal commercial dietary fat inclusions. Interestingly, these two treatments had almost identical measured AME of 13.39 and 13.37 MJ/kg respectively (13.38 and 13.39 MJ/kg AMEn) so no evidence of extra-caloric effects. The widest range of measured AME was 13.07 to 13.49 MJ/kg (treatments 2 and 7 respectively) yet these diets were similar at 62.6 and 62.0 g/kg total crude fat. 

Based on data from Leeson & summers (2005) and van der Klis (2010), the relative energy change is approximately 2% for a change from 59 to 67 g/kg added fat to a diet. So, if soya oil is used at 37 MJ/kg and a 2% decline is used, this will drop to 36.26 MJ/kg at the higher inclusion level. The difference in dietary energy at 67 g/kg inclusion is therefore over-estimated by 0.05 MJ/kg (2.479 versus 2.429 MJ/kg) or 12 kcal/kg feed and this is marginal in our view. The AME system has a number of limitations anyway and it is unclear what the “extra-caloric” values for the small volumes of added soya oil that are questioned could be? 

The small changes in SBM, maize and oil were thus necessary in our view and somewhat novel to other reported work as referenced in the manuscript. 

3) Statistical analysis: The reviewer understands that the high number of treatments required in this design limits capacity for replication. Five replications are low but likely acceptable for the primary objectives of dose response evaluation. However, this replication was likely too low for making means comparisons. Further, the method for separating means should be stated (Tukey’s or other adjustment necessary), and if it was Fisher’s LSD (assumption based on LSD value reported in table), this would be inappropriate given the large number of treatments. Much of the results and conclusions are mean comparisons, however, lack of replication and use of unadjusted mean comparison test are concerning for Type 2 and Type 1 errors, respectively. This reviewer believes that interpretation should only be based on independent and interactive linear and quadratic dose responses and not individual means comparisons.

As noted for reviewer #1 - The Box-Behnken design (BBD) in nutrition research is largely based on de Leon et al., described in reference # [24] of the submitted manuscript. These authors applied a BBD investigating the responses of male broilers to lysine, threonine and total sulphur amino acids from 15 to 35 days post-hatch. In their study, treatments were replicated 5 times based on Myers and Montgomery (1995) recommendations. For this reason, we selected the 5 repeats as being sufficient for a BBD to demonstrate required responses/antagonisms with branched chain amino acids. 

Additionally, in retrospect, a simple one-way ANOVA is not the correct way to do response surface methodology (RSM) and the statistics have been re-run as a BBD in JMP Pro version 15.2.0. 

In summary, to be suitable for publication, this paper should be revised to bring forward and provide reasonable justification for use of such high levels of BCAA (5 to 60% excess of likely requirement), and data interpretation should be based only on dose responses. Given that primary ingredients outside of those tested varied across treatments, responses on dietary AME, N utilization, and AA digestibility should be deemphasized.

In a series of trials in our laboratory, digestive dynamics including AME, N utilisation and AA digestibility have been included. Our contention is that this is important in elucidating treatment effects at the nutrient and metabolic levels. However, we are cognisant that this section can be de-emphasised and will do so. 

The justification for using extreme levels of BCAA in this trial was to ensure that antagonisms reported in the literature could be investigated without having to extrapolate the data beyond the experimental treatments, which is one of the potential pitfalls of RSM. If a dose response existed for BCAA antagonisms, using ratios that were high, were worth investigating in our view. This was novel for this study only because it was a 3-factor BBD. In recent work, Zeitz et al., (2019) investigated ratios of 35 to 60% above the breeder recommendations (references #20 & #21 in the manuscript) so it isn’t completely unusual to test such wide ratios. It would appear that antagonistic effects may only happen in atypical diets that are marginal in Ile and Val.

---

## [Decision Letter · Decision Letter 1]

21 Dec 2021

PONE-D-21-12314R1A multivariate Box-Behnken assessment of elevated branched-chain amino acid concentrations in reduced crude protein diets offered to male broiler chickensPLOS ONE

Dear Dr. Liu,

Thank you for submitting your manuscript to PLOS ONE. After careful consideration, we feel that it has merit but does not fully meet PLOS ONE’s publication criteria as it currently stands. Therefore, we invite you to submit a revised version of the manuscript that addresses the points raised during the review process.

We look forward to receiving your revised manuscript.

Kind regards,

Juan J Loor

Academic Editor

PLOS ONE

Reviewers' comments:

Reviewer's Responses to Questions

**Comments to the Author**

1. If the authors have adequately addressed your comments raised in a previous round of review and you feel that this manuscript is now acceptable for publication, you may indicate that here to bypass the “Comments to the Author” section, enter your conflict of interest statement in the “Confidential to Editor” section, and submit your "Accept" recommendation.

Reviewer #2: All comments have been addressed

2. Is the manuscript technically sound, and do the data support the conclusions?

Reviewer #2: Yes

3. Has the statistical analysis been performed appropriately and rigorously? 

Reviewer #2: Yes

4. Have the authors made all data underlying the findings in their manuscript fully available?

Reviewer #2: Yes

5. Is the manuscript presented in an intelligible fashion and written in standard English?

Reviewer #2: Yes

6. Review Comments to the Author

Reviewer #2: General comments:

This work represents a BCAA BB experiment to assess levels of BCAA in adequacy to excessive level cross product interactions. Indeed, more work with BCAA is needed in broiler nutrition.

The corrections in R1 to remove mean comparisons are very important, as BB designs allow for response surface cross product trend responses, not for the comparison of predicted means (Tables 6 onward are now valid). Some scientists disagree with means being presented in BB studies, just response curves, but I think providing the data allow for future modeling studies. However, why were p values maintained on Tables 15-17, these should be removed?

In general, a lot of the discussion focuses on individual predicted treatments, which is not the intended outcome of the experimental design. Tailoring the discussion to the general trends based on the generated response surfaces or linear and quadratic trends will improve the paper.

Specific comments

L103-114 in R1 I feel more explanation of the experimental design treatment levels is needed. The argument of branched-chain amino acid antagonism being a major issue in reduced CP diets is not synonymous in BCAA excesses, nor industry relevance when feeding reduced CP. Some sort of digestive hypothesis needs to be attempted. For example, and just an example… “Hence, there have been studies conducted testing isoleucine and valine levels at practical levels when leucine is in excess, therefore we tested excess levels of all three in an attempt to determine if allowing for a “balanced” internal ratio allows for correction of the negative effects of leucine.”

In Table 8 there is a column of cross-product p vales, this should be footnoted as to the effect. As BB designs provide three cross-product responses. I feel more explanation to the fitted model is needed. Same in Table 9, footnotes need to be added to explain the data.

L194 Why were only the highest Leu used? And the last treatment does not match Table 15 (10J or 11K). And I feel in all cases of treatment coding (5E-10J and 1-13) in tables, that the actual BCAA levels should be listed to aid in reader interpretation.

L499 in R1, both Maynard et al. [29] and Kidd et al [28] had BCAA carcass products interactions.

7. PLOS authors have the option to publish the peer review history of their article (what does this mean?). If published, this will include your full peer review and any attached files.

Reviewer #2: No

---

## [Author Response · Author response to Decision Letter 1]

2 Mar 2022

This work represents a BCAA BB experiment to assess levels of BCAA in adequacy to excessive level cross product interactions. Indeed, more work with BCAA is needed in broiler nutrition.

The corrections in R1 to remove mean comparisons are very important, as BB designs allow for response surface cross product trend responses, not for the comparison of predicted means (Tables 6 onward are now valid). Some scientists disagree with means being presented in BB studies, just response curves, but I think providing the data allow for future modeling studies. However, why were p values maintained on Tables 15-17, these should be removed?

P-values are removed from tables 15-17

In general, a lot of the discussion focuses on individual predicted treatments, which is not the intended outcome of the experimental design. Tailoring the discussion to the general trends based on the generated response surfaces or linear and quadratic trends will improve the paper.

Specific comments

L103-114 in R1 I feel more explanation of the experimental design treatment levels is needed. The argument of branched-chain amino acid antagonism being a major issue in reduced CP diets is not synonymous in BCAA excesses, nor industry relevance when feeding reduced CP. Some sort of digestive hypothesis needs to be attempted. For example, and just an example… “Hence, there have been studies conducted testing isoleucine and valine levels at practical levels when leucine is in excess, therefore we tested excess levels of all three in an attempt to determine if allowing for a “balanced” internal ratio allows for correction of the negative effects of leucine.”

It now reads like this,

BCAA. The removal of supplemented Val from a balanced reduced crude protein diet caused the largest reduction on growth performance in comparison to removal of others supplemented amino acids and the removal of Leu significantly increased Val and Ile concentrations in plasma [24]. Previous studies examined moderate ranges of BCAA derived from formulating conventional diets [25, 26] and it is hypothesised the importance of BCAA may be more pronouced in reduced CP diets, hence the purpose

In Table 8 there is a column of cross-product p vales, this should be footnoted as to the effect. As BB designs provide three cross-product responses. I feel more explanation to the fitted model is needed. Same in Table 9, footnotes need to be added to explain the data.

The cross-product columns are deleted in both tables 8 and 12. Footnote is not included because Tables 9 and 13 included the significance of each cross-product.

L194 Why were only the highest Leu used? And the last treatment does not match Table 15 (10J or 11K). And I feel in all cases of treatment coding (5E-10J and 1-13) in tables, that the actual BCAA levels should be listed to aid in reader interpretation.

The below is included 

The removal of supplemented Leu increased Val and Ile levels in plasma; whereas the removal of Val and Ile individually did not alter BCAA concentrations in plasma [24]. Therefore, at 27 days post-hatch, three birds at random were selected from each cage of the highest leucine treatments (5E, 7G, 9I and 11K)

L499 in R1, both Maynard et al. [29] and Kidd et al [28] had BCAA carcass products interactions.

Corrected

---

## [Editor Report · Decision Letter 2]

15 Mar 2022

A multivariate Box-Behnken assessment of elevated branched-chain amino acid concentrations in reduced crude protein diets offered to male broiler chickens

PONE-D-21-12314R2

Dear Dr. Liu,

We’re pleased to inform you that your manuscript has been judged scientifically suitable for publication and will be formally accepted for publication once it meets all outstanding technical requirements.

Kind regards,

Juan J Loor

Academic Editor

PLOS ONE
---

## [Editor Report · Acceptance letter]

22 Mar 2022

PONE-D-21-12314R2 

A multivariate Box-Behnken assessment of elevated branched-chain amino acid concentrations in reduced crude protein diets offered to male broiler chickens 

Dear Dr. Liu:

I'm pleased to inform you that your manuscript has been deemed suitable for publication in PLOS ONE. Congratulations! Your manuscript is now with our production department. 

Kind regards, 

on behalf of

Dr. Juan J Loor 

Academic Editor

PLOS ONE